# OCCVLA: VISION-LANGUAGE-ACTION MODEL WITH IMPLICIT 3D OCCUPANCY SUPERVISION

## ABSTRACT

Multimodal large language models (MLLMs) have shown strong vision–language reasoning abilities but still lack robust 3D spatial understanding, which is critical for autonomous driving. This limitation stems from two key challenges: (1) the difficulty of constructing accessible yet effective 3D representations for open-world object modeling, and (2) the loss of fine-grained spatial details in VLMs due to the absence of large-scale 3D vision–language pretraining. To address these challenges, we propose OccVLA, a novel framework that integrates 3D occupancy representations into a unified multimodal reasoning process. Unlike prior approaches that rely on explicit 3D inputs, OccVLA treats dense 3D occupancy as both a predictive output and a supervisory signal, enabling the model to learn fine-grained spatial structures directly from 2D visual inputs. The occupancy prediction are regarded as implicit reasoning processes and can be skipped during inference without performance degradation, thereby adding no extra computational overhead. OccVLA achieves state-of-the-art results on the nuScenes benchmark for trajectory planning and demonstrates superior performance on 3D visual question-answering tasks, offering a scalable, interpretable, and fully vision-based solution for autonomous driving.

## 1 INTRODUCTION

Recently, end-to-end autonomous driving (Hu et al., 2022; Jiang et al., 2023; contributors, 2023; Hu et al., 2023; Xie et al., 2025) has witnessed remarkable advances, driven by increasing demands for real-world deployments. Advanced autonomous driving systems (Zhou et al., 2025a; Zheng et al., 2025) now routinely integrate vision language models (VLMs) to deliver compelling reasoning capabilities in complex driving scenarios. Nevertheless, the persistent gap between 2D and 3D perception remains a principal limitation to broader VLM adoption. In autonomous driving, robust 3D perception (Qi et al., 2017; Lang et al., 2019; Wang et al., 2022) is indispensable for localization and navigation, since its fidelity directly influences the safety of downstream decision-making. Prior work has extensively explored this challenge as shown in Fig. 1 (a). In VLM-based perception pipelines (Tian et al., 2024; Hwang et al., 2024), supervision relies on 3D annotations described in text (e.g., coordinates or bounding boxes), which are inherently sparse. Moreover, Coordinates and bounding boxes can only model predefined object categories and lack geometric details of objects (Tian et al., 2023), thereby limiting their 3D representation capability. As illustrated in Fig. 1 (b), some recent methods (Wang et al., 2025; Wei et al., 2024; Xiong et al., 2023) attempt to incorporate 3D inputs, but they are limited by the lack of large-scale 3D vision–language pretraining and detailed captions for complex spatial scenes. Such 3D VLMs generally focus on supervising textual outputs while overlooking the rich 3D visual modality, leaving potential for improving spatial understanding in autonomous driving.

Two critical challenges arise in this context: (1) establishing an accessible and effective representation of 3D information, and (2) developing dense 3D supervision to preserve fine-grained spatial details. Recent progress in automated annotation pipelines (Tian et al., 2023; Ye et al., 2025) enables large-scale acquisition of 3D occupancy representations for autonomous driving scenarios. Such representations naturally encode both detailed 3D structural geometry and semantic labels, providing a unified format for aligning spatial and semantic information. With advancements in occupancy prediction techniques, transformer-based models (Li et al., 2023b; Huang et al., 2023; Zhang et al., 2023) have demonstrated their feasibility for modeling this representation. Inspired by

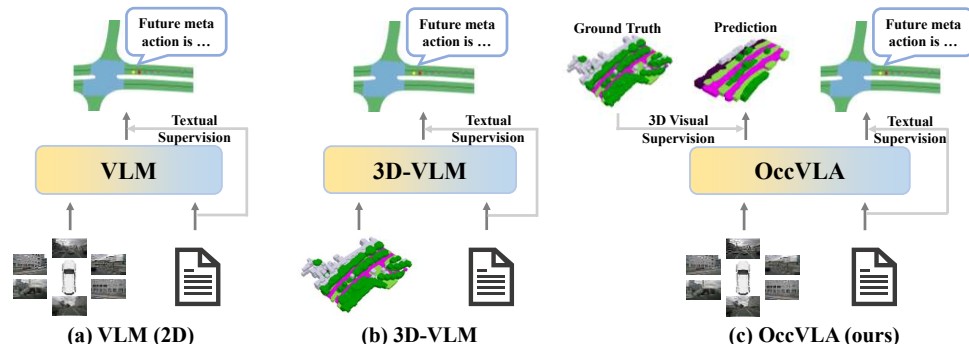

Figure 1: Comparison of autonomous driving VLM architectures. (a) VLM (2D): Takes only 2D visual inputs and relies solely on textual supervision, lacking explicit 3D spatial grounding. (b) 3D-VLM: Consumes explicit 3D inputs (e.g., Occupancy, LiDAR) for reasoning, but the absence of large-scale 3D vision–language pretraining often leads to loss of fine-grained spatial details and limits generalization. (c) OccVLA (ours): Predicts dense 3D occupancy from 2D images and uses it as both an output and a dense 3D supervisory signal, enhancing fine-grained spatial understanding while preserving rich 2D visual details.

these developments (Li et al., 2023c;a), we propose a VLM augmented with occupancy prediction capabilities, to simultaneously address the representation and supervision challenges.

Building on this perspective, we introduce a novel framework, **Occ**upancy **V**ision-**L**anguage-**A**ction model (OccVLA), which enables execution of occupancy prediction, vision-language reasoning and action generation. As illustrated in Fig. 2, OccVLA treats occupancy tokens as implicit reasoning processes, using cross-attention to receive visual features from intermediate layers of the VLM. To address the spatial sparsity of occupancy representations (Wei et al., 2024), we first predict occupancy in a compact latent space, after which an occupancy head maps the resulting occupancy tokens back to the high-resolution original occupancy space. This 3D scene prediction step enables the VLM to capture fine-grained spatial details more effectively. Moreover, compared to raw visual features, supervising on the occupancy representation substantially enhances the 3D representational capacity of the VLM's visual features. Notably, during inference, the occupancy prediction process can remain inactive, introducing no additional computational overhead. Finally, a lightweight MLP consumes the meta-actions predicted by the VLM to predict future trajectories, providing a simple yet effective solution for trajectory forecasting. To facilitate robust training of this action-generation module, we construct an automated data pipeline to generate meta-action labels, 3D occupancy, and corresponding Chain-of-Thought (CoT) reasoning annotations.

OccVLA demonstrates superior performance across multiple perception and planning tasks. We further validate its 3D understanding capabilities on the NuScenes-QA benchmark(Qian et al., 2023), such as relative vehicle position localization. The visual input to OccVLA consists of only 2D images, which effectively preserves the inherent generalization capability of VLMs during open-domain dialogue. Notably, OccVLA offers the flexibility to decode the occupancy representation, producing interpretable and quantitatively evaluable outputs, which is particularly advantageous for fully vision-based autonomous driving solutions.

The main contributions of this paper are as below:

1. We propose the autonomous driving framework OccVLA, which extends the 3D reasoning capabilities of vision-language models (VLMs) through the occupancy prediction process while effectively preserving visual information from 2D images.

2. The design of the cross-modal attention allows the model to skip the occupancy prediction process during inference, introducing no additional computational complexity.

3. OccVLA achieves outstanding performance in both end-to-end trajectory planning and 3D VQA tasks, setting state-of-the-art results on the public benchmark nuScenes.

4. We construct OccVLA-nuScenes, a new dataset derived from nuScenes. This dataset includes the meta-action, 3D occupancy, and Chain-of-Thought (CoT) training data used in our motion planning experiments, which we will make public to encourage further research.

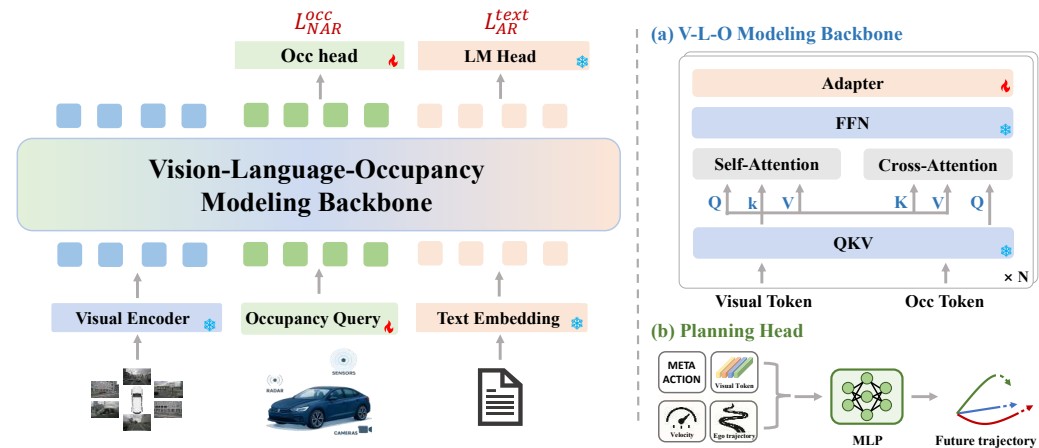

Figure 2: Overview of the proposed OccVLA architecture. The framework unifies dense 3D occupancy (occ) prediction and language modeling within a shared Vision–Language–Occupancy (V-L-O) backbone. The model is jointly trained with $L_{NAR}^{occ}$ (a non-autoregressive loss for occupancy prediction) and $L_{AR}^{text}$ (an autoregressive loss for textual outputs). (a) In the V-L-O backbone, occupancy tokens query visual features from visual tokens through cross-attention, while visual tokens are modeled via self-attention. (b) After predicting meta actions through the VLM, a planning head (MLP) generates the future trajectory.

## 2 RELATED WORK

### 2.1 MLLMS IN AUTONOMOUS DRIVING

Recent studies (Sima et al., 2023; Wang et al., 2023; Zhang et al., 2025) argue that multimodal large language models (MLLMs) can emulate the human thought process during driving. Leveraging the exceptional zero-shot generalization capabilities of vision-language models (VLMs) (Tian et al., 2024; Xu et al., 2024), they can effectively handle long-tail scenarios in autonomous driving. However, due to limitations in their pretraining paradigms, VLMs struggle to effectively comprehend the 3D structure of the physical world. DriveVLM (Tian et al., 2024) is the first to propose using VLMs for autonomous driving motion planning, but it relies on high-quality annotated datasets. EMMA (Hwang et al., 2024) utilizes large-scale datasets containing 3D coordinates to enhance the model's 3D grounding capability, but this approach remains incapable of performing fine-grained modeling of open-world 3D scenes. Similarly, OmniDrive (Wang et al., 2025) compresses 3D point clouds into sparse queries and feeds them into large language models (LLMs), which necessitates additional 3D sensors and forces the model to process large-scale 3D inputs. In this work, we propose OccVLA, which leverages auto-annotation occupancy data to provide dense 3D supervision for MLLMs.

### 2.2 OCCUPANCY FOR 3D PERCEPTION

3D occupancy assigns semantic labels to spatial grids, aiming to establish fine-grained representations of 3D scenes. Transformer-based methods (Liu et al., 2024b; Li et al., 2024a), through spatiotemporal feature fusion, have demonstrated significant advantages in occupancy prediction tasks. Recently, unlike traditional vision-language models (VLMs), several studies have explored the potential of using occupancy as input of LLM to enhance the understanding capabilities of multimodal large language models (MLLMs) in autonomous driving. OccWorld (Zheng et al., 2024) proposes making predictions on multi-scale occupancy features to learn a world model, while OccLLAMA (Wei et al., 2024) introduces the use of large language models (LLMs) to predict future 3D occupancy and actions. Similarly, Occ-LLM (Xu et al., 2025b) proposes a motion-separating variational autoencoder that disentangles dynamic and static objects in occupancy grids and predicts them separately using LLMs. Although it is possible to perform joint training of 3D visual inputs and language similar to VLMs, there remains a risk that captions omit critical 3D information. To address these limitations, OccVLA focuses on using occupancy as both the model's output and supervision signal, thereby establishing a novel framework for multimodal learning.

## 3 METHOD

### 3.1 OVERVIEW

In this section, We propose OccVLA, a unified framework for 3D occupancy prediction and future ego-motion planning. The core components of OccVLA include the occupancy prediction (Section 3.2) and an independent planning head (Section 3.3). Additionally, we introduce a three-stage training process (Section 3.4) to better balance the model's performance across different tasks.

We incorporate 3D visual supervision into the typical VLM framework, as illustrated in Fig. 2. Before performing next-token prediction, the model first perceives the visual input and produces an occupancy prediction. This unified architecture enables seamless integration of visual and textual information during the perception stage (perceive first, then reason), thereby establishing a solid perceptual foundation for visual understanding, mitigating the information loss caused by text-only supervision, and ultimately enhancing the model's 3D comprehension capability.

### 3.2 OCCUPANCY PREDICTION

**Occupancy Transformer.** To strengthen the 3D perception capability of autonomous driving systems, we extend the original VLM framework with a dedicated 3D occupancy prediction processing. OccVLA takes a set of learnable occupancy queries as input, which are passed through the same feed-forward layers, query–key–value (QKV) projections, and normalization layers as in the VLM. Cross-modal interaction is enabled through a shared visual key–value (KV) representation, which allows the occupancy tokens to query visual features. As illustrated in Fig. 2(a), the occupancy tokens (right) can access visual features (left) from the vision–language model via cross-attention. We can formally describe the attention operations as follows:

$$h_O^{occ} = O(softmax(\frac{h_Q^{occ}[h_K^{img}]^T}{\sqrt{d}})[h_V^{img}]) \tag{1}$$

$$h_O^{img} = O(softmax(\frac{h_Q^{img}[h_K^{img}]^T}{\sqrt{d}})[h_V^{img}]) \tag{2}$$

where $h_O^{img}$ denotes the image features output by the left-side of VLM, while $h_O^{occ}$ denotes the occupancy features generated by the right-side of model. Here, $h_Q, h_K$ and $h_V$ are the query, key, and value representations, and $O$ is unified output projections. Empirically, for the text reasoning process, we observe that whether text tokens have access to occupancy features does not result in a significant difference in quality after model convergence. This suggests that text can be predicted solely from visual features, indicating that during language inference, additional occupancy computation is unnecessary, thereby improving efficiency. Finally, We insert lightweight adapters at the residual connections to fintune the VLM and preserve the original vision–language modeling capabilities.

**Latent Occupancy Prediction.** In autonomous driving scenarios, approximately 90% of the 3D space is empty (Wei et al., 2024), resulting in highly sparse occupancy signals. Moreover, the raw occupancy grid is memory-intensive, typically represented as $x \in R^{H \times W \times D}$ with $(H, W, D) = (200, 200, 16)$ (Tian et al., 2023), making direct prediction inefficient. We follow Zheng et al. (2024), mapping the target occupancy to a compact latent space $y \in R^{\frac{H}{r} \times \frac{W}{r} \times F}$, where r is downsampling rate and F is the feature dimension of latents. As illustrated in Fig. 2, the left-side occupancy model outputs hidden states $h_O^{occ}$, which are projected into $z \in R^{\frac{H}{r} \times \frac{W}{r} \times F}$ via a linear projector. These features are then fed into the VQ-VAE decoder which is initialized with pretrained weights from Zheng et al. (2024). Finally, a classification head converts the decoded features into the 3D occupancy predictions.

### 3.3 MOTION PLANNING

**Task Decomposition.** Large Language Models (LLMs) and Vision-Language Models (VLMs) excel at reasoning over semantic cues, but exhibit limited sensitivity to precise numerical values . Directly

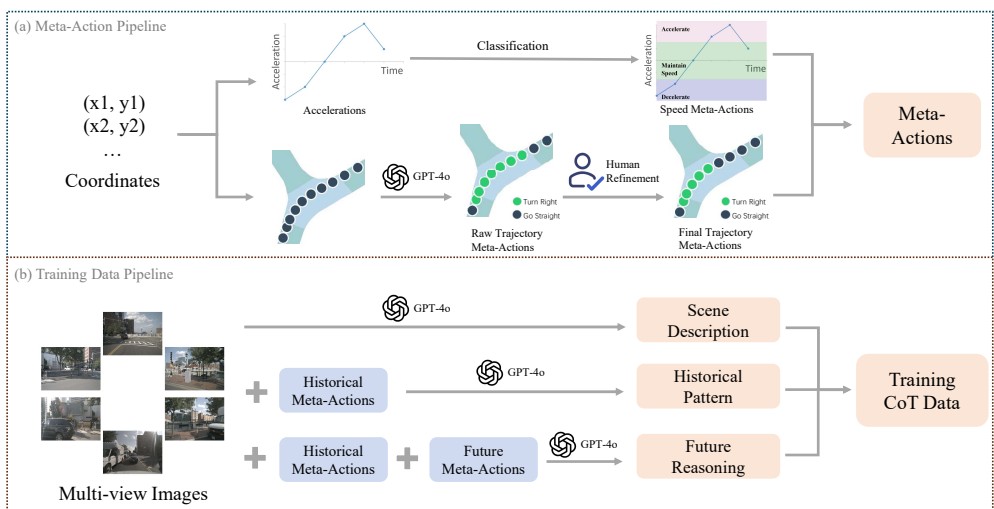

Figure 3: Overview of the meta action and CoT data generation pipeline. (a) Meta Action Pipeline: Vehicle trajectory coordinates are processed to compute accelerations for velocity action classification, and matched to HD map lanes for trajectory action classification via GPT-4o, followed by human refinement. The two components are combined to produce final meta actions. (b) Training Data Pipeline: Multi-view images and meta actions are provided to GPT-4o to generate scene descriptions, infer historical motion patterns, and perform future reasoning, forming CoT training data.

predicting future vehicle coordinates from raw trajectories therefore underutilizes their strengths. Following Tian et al. (2024), we decompose motion planning into two stages: (1) predicting a high-level *meta action* in natural language form, and (2) generating precise future coordinates using a lightweight model conditioned on the predicted meta actions.

**Meta Action Prediction.** We define a *meta action* as a compact, interpretable representation of the vehicle's short-term driving intent, consisting of two orthogonal components: (1) *velocity action*, categorized into *Maintain speed*, *Accelerate*, and *Decelerate*; and (2) *directional action*, categorized into *Go Straight*, *Turn Left*, *Turn Right*, *Bear Left*, *Bear Right*, and *Stop*. This formulation allows the model to reason in a discrete, language-friendly space while retaining key motion semantics.

To better utilize the reasoning capabilities of large language models, we follow Hwang et al. (2024) and construct chain-of-thought (CoT) supervision for meta action prediction. The input to the VLM consists of six images captured from multiple perspectives, along with the past meta actions of the ego vehicle. The model first generates a natural language description of the scene, then infers the driver's intent based on historical meta actions, and finally outputs the predicted future meta action. This multi-step reasoning encourages the model to explicitly connect scene understanding with motion intent prediction.

We develop a fully automated data construction pipeline to generate both meta action labels and their corresponding CoT annotations of our OccVLA-nuScenes dataset. For the velocity component, labels are directly obtained via threshold-based classification on acceleration. For the directional component, future trajectories are projected onto a lane-level HD map and classified by GPT-4o (OpenAI et al., 2024) into one of the five directional categories. For the CoT annotations, GPT-4o is prompted to produce scene descriptions based on the image inputs, and then, given the ground truth meta action, to complete the reasoning steps leading to the correct label.

To ensure annotation quality, all generated meta actions are manually inspected, and about 20% percent of the data has been further refined to achieve better consistency. Since the BEV perspective enables simultaneous inspection of all trajectory coordinates in a scene, minimal manual annotation effort is required. Fig. 3 demonstrates our meta action and training data pipeline.

**Planning Head.** Given the predicted meta action, the planning head translates this high-level intent into concrete future coordinates. We adopt a simple MLP architecture inspired by (Li et al., 2024b), taking as input the meta action embedding, the previous timestep velocity, and visual tokens from the VLM. The model predicts the vehicle's position for the next 3 seconds. Notably, no high-level

Figure 4: Overview of the training pipeline. Stage 1: Pretraining the VLM on autonomous driving scenarios using visual and text inputs. Stage 2: Occupancy-language joint training to enhance 3D scene understanding. Stage 3: Planning head training where the planning head predicts future coordinates from VLM-generated meta actions.

navigation commands are provided, ensuring that all planning decisions emerge solely from the model's scene understanding.

## 3.4 TRAINING STAGE

**Pretraining in Autonomous Driving Scenarios.** As shown in Fig 4, we we adopt a VLM fine-tuning strategy along with its corresponding loss functions using the dataset sampled from OmniDrive(Wang et al., 2025). This phase helps the model transfer from general domains to autonomous driving scenarios, such as focusing on specific types of objects (e.g., cars, pedestrians, roads, etc.) or predicting future motion. Additionally, this training approach prepares the model to perform long-text reasoning and engage in dialogue, making it more effective in handling complex language understanding tasks.

**Occupancy-Language Joint Training.** We focus on improving the 3D understanding capability of the VLM by aligning the Occupancy-vision modality during training. The full Occupancy-image-language dataset is used to supervise the model training, with the former eliciting 3D information representation from visual features, while the latter ensures consistency in 3D scene descriptions. To leverage the deep features of the model, we apply adapters (Pfeiffer et al., 2020; Poth et al., 2023) to fine-tune the transformer blocks. We combine the standard autoregressive language modeling loss of the LLM, $\mathcal{L}_{ce}^{text}$ with a non-autoregressive 3D perception loss, $\mathcal{L}_{ce}^{occ}$, which calculate the cross-entropy between predicted occupancy logits and ground-truth occupancy labels. We observe that directly aligning the latent space features is suboptimal due to the inherent biases introduced by VQ-VAE encoding. Therefore, we choose to directly supervise the final 3D occupancy categories. Following (Shi et al., 2025), we adopt separate learning rates for different modules to further enhance training stability: the VQ-VAE decoder is assigned a learning rate of zero (rather than being fully frozen) to maintain gradient flow, while all other components share a common learning rate.

$$\mathcal{L} = \mathcal{L}_{AR}^{text} + \lambda_{occ}\mathcal{L}_{NAR}^{occ} \tag{3}$$

where $\lambda_{occ}$ is a factor that controls the degree of focus on occupancy.

**Planning Head Training.** To address the trajectory planning task, the planning head takes as input the meta actions predicted by the VLM, along with current velocity, visual tokens from the output of vlm and ego trajectories, and outputs the coordinates of the future trajectory. Specifically, the meta actions predicted by the trained VLM are fed into the planning head, whose outputs are supervised using a mean squared error (MSE) loss computed against the ground-truth trajectory coordinates.

## 4 EXPERIMENT

### 4.1 EXPERIMENT SETTINGS

**Dataset** NuScenes is a widely used dataset in autonomous driving, consisting of 700 training scenes and 150 validation scenes. Based on the sensor information (such as images and radar) in NuScenes, Occ3D is developed as a large-scale dataset representing 3D occupancy. Furthermore, in recent years, with the advancement of large autonomous driving models, many Visual Question Answering (VQA) datasets have been built on NuScenes. We specifically evaluate the model's capabilities in 3D localization, object querying, and relational comparison using NuScenes-QA (Qian et al., 2023). Additionally, we collect a large-scale image-occupancy-text dataset to align multiple modalities and train the model to predict future meta-actions. This multimodal alignment and future prediction task

Table 1: End-to-end motion planning experiments on nuScenes Caesar et al. (2020) with different input and supervision. L denotes LiDAR input and C denotes camera input.

| Method | Input | Supervision | L2(m)↓ | | | |
|--------|-------|-------------|------|------|------|------|
| | | | 1s | 2s | 3s | Avg. |
| NMP | L | Box & Motion | 0.53 | 1.25 | 2.67 | 1.48 |
| FF | L | Freespace | 0.55 | 1.20 | 2.54 | 1.43 |
| ST-P3 | C | Map & Box & Depth | 1.33 | 2.11 | 2.90 | 2.11 |
| UniAD | C | Map & Box & Motion & Track & Occ | 0.48 | 0.96 | 1.65 | 1.03 |
| VAD | C | Map & Box & Motion | 0.54 | 1.15 | 1.98 | 1.22 |
| DriveVLM-Dual | C | Map & Box & Motion | 0.15 | 0.29 | 0.48 | 0.31 |
| EMMA | C | None | **0.14** | 0.29 | 0.54 | 0.32 |
| OmniDrive | C & L | None | **0.14** | 0.29 | 0.55 | 0.33 |
| S4-Driver | C | None | 0.16 | 0.34 | 0.63 | 0.38 |
| Ours | C | Occ | 0.18 | **0.26** | **0.40** | **0.28** |

Table 2: End-to-end motion planning experiments on nuScenes Caesar et al. (2020) compared with models like OccNet Liu et al. (2024a), OccWorld Zheng et al. (2024), and others that use occupancy as LLM input.

| Method | Input | Supervision | L2(m)↓ | | | |
|--------|-------|-------------|------|------|------|------|
| | | | 1s | 2s | 3s | Avg. |
| OccNet | Occ | Map & Box | 1.29 | 2.31 | 2.98 | 2.25 |
| OccWorld-O | Occ | None | 0.43 | 1.08 | 1.99 | 1.17 |
| OccLLAMA-O | Occ | None | 0.37 | 1.02 | 2.03 | 1.14 |
| Occ-LLM | Occ | None | **0.12** | **0.24** | 0.49 | **0.28** |
| OccWorld-F | C | Occ | 0.45 | 1.33 | 2.25 | 1.34 |
| OccLLama-F | C | Occ | 0.38 | 1.07 | 2.15 | 1.20 |
| Occ-LLM | C | Occ | 0.21 | 0.40 | 0.67 | 0.43 |
| Ours | C | Occ | 0.18 | 0.26 | **0.40** | **0.28** |

aim to enhance the model's understanding of 3D scenes and its ability to reason about and act within dynamic autonomous driving scenarios.

**Implementation Details** For all experiments, we adopt the Paligemma2-3B-224px (Beyer et al., 2024; Steiner et al., 2024) as the vision-language model backbone, while the scene VQVAE is initialized following the settings in OccWorld (Zheng et al., 2024). We train all models using AdamW (Loshchilov & Hutter, 2019) optimizer, and conduct experiments on 8× NVIDIA A800 GPUs.

## 4.2 RESULTS AND ANALYSIS

**Motion Planning** As shown in Table 1, we compare the motion planning capabilities of OccVLA with several strong baselines that utilize various inputs and supervisions. We observe that the current state-of-the-art method, EMMA(Hwang et al., 2024), relies on supervision annotations (3D/BEV coordinates & 3D bounding box), which limits its scalability to large-scale datasets. OmniDrive(Wang et al., 2025), on the other hand, depends on inputs from both camera and lidar. In contrast, OccVLA requires only camera input and uses occupancy, which can be annotated at scale, as supervision. We achieve state-of-the-art performance in terms of average L2 distance and competitive results in trajectory planning within 3 seconds.

In Table 2, methods like Occ-LLM, which use occupancy as input to the LLM, encode strong 3D priors and achieve superior performance across multiple metrics. These methods use camera input and obtain Occupancy through an occupancy prediction network before feeding it into the LLM. Our method directly takes camera input and integrates the Occupancy prediction process into the LLM, achieving state-of-the-art results. Excitingly, OccVLA achieves competitive performance using only camera input compared to methods that use ground-truth Occupancy as input, further highlighting the advantage of using occupancy as an LLM output. Additionally, we achieve better performance

Table 3: Quantitative results on Nuscenes-QA(Qian et al., 2023) compared with models that using different input like LLAVA (Liu et al., 2023), LiDAR-LLM(Yang et al., 2023), OccLLaMA(Wei et al., 2024) and OpenDriveVLA(Zhou et al., 2025b).

| Model | Size | Input | exist(%)↑ | | | count(%)↑ | | | object(%)↑ | | | status(%)↑ | | | comparison(%)↑ | | | acc(%)↑ |
|---|---|---|---|---|---|---|---|---|---|---|---|---|---|---|---|---|---|---|
| | | | h0 | h1 | all | h0 | h1 | all | h0 | h1 | all | h0 | h1 | all | h0 | h1 | all | |
| LLaVA | 7B | C | 74.8 | 72.9 | 73.8 | 14.9 | 14.3 | 14.6 | 57.7 | 34.5 | 37.9 | 48.6 | 44.5 | 45.9 | 65.9 | 52.1 | 53.3 | 47.4 |
| LiDAR-LLM | 7B | L | 79.1 | 70.6 | 74.5 | 15.3 | 14.7 | 15.0 | 59.6 | 34.1 | 37.8 | 53.4 | 42.0 | 45.9 | 67.0 | 57.0 | 57.8 | 48.6 |
| OccLLaMA3.1 | 8B | Occ | 82.9 | 79.2 | 80.9 | 19.2 | 19.2 | 19.2 | 64.8 | 43.1 | 46.3 | 51.0 | 46.1 | 47.8 | 76.5 | 65.6 | 66.6 | 54.5 |
| OpenDriveVLA | 7B | C | - | - | 84.2 | - | - | 22.7 | - | - | 49.6 | - | - | 54.5 | - | - | 68.8 | 58.2 |
| Ours | 3B | C | 87.4 | 81.7 | 84.3 | 22.6 | 21.2 | 21.9 | 73.6 | 51.2 | 54.5 | 62.6 | 57.9 | 59.5 | 79.2 | 66.0 | 67.2 | 59.5 |

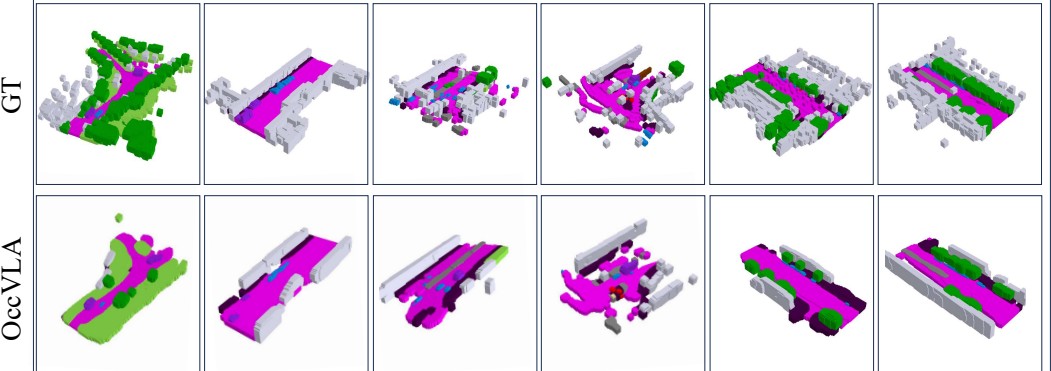

Figure 5: The 3D occupancy forecasting results of our OccVLA, which demonstrate accurate estimation for critical objects (e.g., vehicles, roads, etc.).

than OccLLaMA (7B) Wei et al. (2024); Touvron et al. (2023) with only a 3B model, demonstrating greater potential for practical applications.

**Visual Question Answering** To further evaluate the 3D understanding capability of our model, we test it on the challenging NuScenes-QA (Qian et al., 2023) benchmark. The NuScenes-QA dataset is specifically designed for autonomous driving scenarios, providing 460,000 question-answer pairs. The questions cover diverse types including existence, counting, object and status queries, and comparisons, designed to test a model's reasoning in intricate street views. Table 3 shows the overall accuracy on NuScenes-QA. By incorporating occupancy supervision, our 3B-parameter, image-only VLM successfully outperforms larger models that rely on 3D inputs from LiDAR or explicit ground-truch occupancy data. This result highlights the superiority of our approach in fostering a deeper and more efficient 3D understanding from visual-only inputs in autonomous driving.

**Occupancy Prediction** The goal of this task is to predict real-time 3D occupancy using multi-view images captured by cameras. Although we employ an LLM-based architecture that is not specifically designed for occupancy prediction, our model demonstrates competitive performance, outperforming baseline methods. Specifically, the model processes only the current time-step input without leveraging features from past states and directly outputs the 3D occupancy for the current moment, achieving about 10% in the mIoU metric. As illustrated in the Fig. 5, the absence of multi-timestamp image inputs predictably limits the model's ability to handle occluded regions (e.g., buildings hidden behind trees). Nevertheless, the model excels at predicting key elements in autonomous driving scenarios, such as lanes, vehicles, pedestrians, and finer details of objects in proximity to the vehicle. Therefore, the model exhibits a strong object-level understanding of 3D scenes in the context of autonomous driving. Despite the lack of temporal information, it effectively leverages multi-view images from the current time step to produce high-quality 3D occupancy predictions. This highlights the potential of LLM-based architectures in such tasks, even though they are not originally designed for this purpose.

## 4.3 ABLATION STUDY

**Occupancy Supervision.** We compare the impact of the occupancy prediction process on the performance of both motion planning and visual question answering (VQA) tasks. As shown in the table 4, in the absence of occupancy supervision, the model can only rely on its understanding of 2D images to plan future actions. As illustrated in Fig. 6, when processing multi-view images, the model

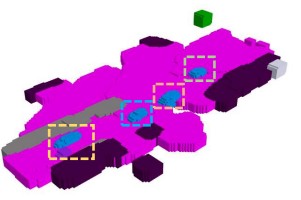

(a) Multi-view Images (b) Predicted Occupancy

**Training w/ Occupancy:**
In cloudy afternoon, the vehicle is at a city intersection surrounded by modern high-rise buildings. Ahead, a green traffic light signals the right of way, while the traffic light is green, allowing vehicles to proceed. To the right-front, **a white SUV** is parked near a sidewalk lined with trees and a parking structure… Behind, a line of vehicles, including **a black SUV** and **a silver sedan**, waits at the light…

**Training w/o Occupancy:**
In cloudy afternoon, At an urban intersection, the traffic light ahead is green, allowing vehicles to proceed. To the right-front, **a white pickup truck** is parked near a sidewalk bordered by modern office buildings. Directly in front, **a black sedan** waits at the intersection… Behind the vehicle, a line of vehicles, including **a black Jeep** and **a black sedan**, follow closely in adjacent lanes. To the left-rear, **a white pickup truck** is parked near the cur…

Figure 6: Scene descriptions generated by OccVLA, where bounding boxes in different colors indicate distinct instances with corresponding image and occupancy predictions. Red text denotes correctly described critical objects, while blue text indicates incorrectly described ones.

Table 4: Ablation study of the occupancy supervision. The ✗ indicates that the model corresponds to the original VLM without occupancy integration, whereas the ✓ denotes that the model is trained through joint occupancy–vision–language learning.

Table 5: Ablation study on Ego Trajectory. The ✗ symbol denotes that the model has no access to Ego Trajectory information.

| Method | Occupancy | Speed (%) | Trajectory (%) | Avg. (%) | Overall. (%) |
|--------|-----------|-----------|----------------|----------|--------------|
| OccVLA | ✗ | 53.77 | 77.24 | 65.51 | 41.48 |
| OccVLA | ✓ | 54.83 | 77.95 | 66.39 | 43.08 |

| Method | Ego Trajectory | L2(m)↓ | | |
|--------|----------------|--------|--------|--------|
| | | 1s | 2s | 3s |
| OccVLA | ✗ | 0.28 | 0.35 | 0.80 |
| OccVLA | ✓ | 0.18 | 0.26 | 0.40 |

struggles to comprehend the spatial relationships between surrounding vehicles, which can lead to hallucinations. In contrast, OccVLA incorporates occupancy supervision, providing the model with additional 3D information that enables it to go beyond sparse textual supervision and enhance its 3D understanding through the process of 3D occupancy prediction. This improvement can be attributed to the occupancy supervision, which strengthens the 3D priors within the visual features learned by the LLM. Consequently, this enhancement leads to approximately a 1.5% improvement in meta-action prediction performance.

**Ego Trajectory.** For motion planning task, previous works (Zhai et al., 2023; Li et al., 2024b) have raised concerns that ego trajectory might introduce excessive priors into the model, potentially leading to overfitting on the dataset. To ensure a fairer comparison, we report planning performance without past trajectory information in the table 5. Under the same conditions, our method demonstrates competitive performance advantages compared to state-of-the-art approaches (e.g., VAD, etc.). This highlights the strong performance and generalization capability of OccVLA, further supporting its effectiveness in diverse scenarios.

## 5 CONCLUSION

In this paper, we propose OccVLA, a novel occupancy-vision-language framework for autonomous driving. OccVLA employs a parallel LLM architecture in the latent space to jointly learn occupancy and vision-language representations. This framework leverages pre-trained 2D knowledge while achieving a more critical fine-grained understanding of 3D spatial semantics. Our approach does not rely on additional 3D input information and can bypass the occupancy prediction process during inference, effectively addressing the inference delay in previous 3D VLM-based autonomous driving models. This paradigm also holds substantial potential for broader applications, such as extending the prediction to future 3D states, which could integrate world model knowledge into autonomous driving MLLMs.

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

## A    ADDITIONAL ABLATION STUDIES

**Ablation on $\lambda_{occ}$.** We further conducted an ablation study to analyze the effect of the occupancy supervision weight $\lambda_{occ}$ as shown in Table 6.

We observe that when $\lambda_{occ} = 0.1$, the effect of occupancy supervision is marginal, producing results close to those without occupancy. When $\lambda_{occ} = 2$, the model tends to overfit to occupancy supervision, leading to degraded overall performance. Therefore, $\lambda_{occ} = 1$ achieves the best balance between leveraging occupancy supervision for improved 3D understanding and avoiding overfitting.

**Ablation on occupancy and COT supervision.** To better illustrate the independent contributions of each component, we conducted a detailed ablation study by selectively removing different supervision signals and analyzing the corresponding performance changes.

As shown in the Table 7, removing both CoT and occupancy supervision leads to a significant performance drop. Introducing CoT supervision alone improves trajectory and speed prediction accuracy, indicating that structured reasoning benefits planning tasks. When CoT and occupancy supervision are jointly applied, the model achieves the best overall performance. This suggests that occupancy supervision enhances spatial understanding of driving scenes while complementing CoT-based reasoning. We also tested a setting where only occupancy supervision was used without CoT. This configuration led to overfitting to the occupancy signal, causing repetitive outputs during autoregressive generation. The baseline model here corresponds to training without the OccVLA-nuScenes dataset, i.e., without occupancy or CoT supervision, relying solely on meta-action training of the VLM model.

## B    EFFECT OF OCCUPANCY TOKENS ON TEXT GENERATION

To provide a more detailed explanation of how OccVLA skips the occupancy prediction process during inference, we present two training settings and evaluate the model's performance under two different configurations regarding the use of occupancy tokens during text generation..

**Setting 1: Text generation with occupancy tokens.**    In this configuration, text generation is conditioned on both image tokens and occupancy tokens. The model explicitly predicts occupancy maps and incorporates them into the reasoning process:

$$P(Y|I,O) = \prod_{t=1}^{T} P(y_t \mid y_{<t}, I, O),  \tag{4}$$

where $I$ denotes image tokens, $O$ represents occupancy tokens, and $Y$ is the target text sequence.

**Setting 2: Text generation without occupancy tokens.**    In this configuration, text generation is conditioned only on image tokens. During training, occupancy supervision is still applied to update model parameters through cross-attention with image tokens. However, during inference, since text generation is not conditional on occupancy tokens, the model can skip the occupancy prediction step without affecting the performance of meta-action prediction:

$$P(Y|I) = \prod_{t=1}^{T} P(y_t \mid y_{<t}, I).  \tag{5}$$

**Experimental results.**    We trained and evaluated the model under both settings. The results are summarized in Table 8.

Table 6: Ablation study of the $\lambda_{occ}$.

| $\lambda_{occ}$ | Speed (%) | Trajectory (%) | Avg. (%) | Overall. (%) |
|---|---|---|---|---|
| 0 | 53.77 | 77.24 | 65.51 | 41.48 |
| 0.1 | 53.85 | 75.77 | 64.81 | 41.45 |
| 1 | **54.83** | **77.95** | **66.39** | **43.08** |
| 2 | 54.01 | 74.07 | 64.04 | 40.64 |

Table 7: Ablation study of the occupancy and COT supervision. The ✗ indicates that the model is trained without occupancy or COT supervision.

| COT | Occupancy | Speed (%) | Trajectory (%) | Avg. (%) | Overall. (%) |
|---|---|---|---|---|---|
| ✗ | ✗ | 53.05 | 76.25 | 64.65 | 40.26 |
| ✗ | ✓ | - | - | - | - |
| ✗ | ✗ | 53.77 | 77.24 | 65.59 | 41.48 |
| ✓ | ✓ | **54.83** | **77.95** | **66.39** | **43.08** |

**Analysis.** Setting 2 achieves approximately a 25% reduction in inference time and even yields a slight performance improvement of about 0.5%. We hypothesize that excessive visual tokens (i.e., occupancy tokens) may introduce redundant information that interferes with reasoning, leading to minor degradation. These findings indicate that our approach achieves both efficiency and accuracy advantages during inference. In this paper, we report all experimental results based on Setting 2, under which **OccVLA** supports flexible inference modes: users can either skip occupancy prediction for faster inference or retain it to visualize the model's spatial understanding of 3D scenes. Overall, this skipping mechanism serves as an optional acceleration strategy rather than a default configuration, and it does not introduce significant discrepancies between training and inference. The model demonstrates robust and consistent performance across both modes.

## C EXTENDED EXPERIMENTS ON WOD-E2E

We evaluate the generalization and scalability of OccVLA on the Waymo Open Dataset End-to-End Driving (WOD-E2E) (Xu et al., 2025a) dataset.

**Dataset Construction:** the WOD-E2E dataset primarily targets long-tail driving scenarios and contains approximately 400k training samples, each with 8 camera views and ego trajectories. However, it lacks point cloud data and occupancy annotations. To construct the training data required by OccVLA, we performed the following labeling steps:

**Meta-action Annotation:** Following the definition in our paper, we decompose the meta action into trajectory (Turn Left, Turn Right, Bear Left, Bear Right, Go Straight) and speed (Stop, Accelerate, Decelerate, Maintain speed). The trajectory labels are obtained via clustering, while speed labels are derived based on acceleration thresholds. The final meta action includes both the overall trajectory direction and the second-by-second speed actions over a 5-second horizon.

**COT Annotation:** After obtaining meta actions, we used our proposed method in conjunction with GPT-4o to generate COT reasoning annotations.

**Occupancy Pseudo-labeling:** Since LiDAR data is unavailable, the original automatic labeling pipeline cannot be applied. We therefore utilized the SurroundOcc(Wei et al., 2023) model (pre-trained on nuScenes) to predict occupancy labels. Most existing occupancy models are designed for 6-camera datasets such as nuScenes, while WOD-E2E provides 8 cameras. To ensure compatibility, we first align and stitch five front-facing images (side-left, front-left, front, front-right, side-right) using the provided camera intrinsics and extrinsics. We then recrop the stitched view back into three image views, new-front-left, new-front, and new-front-right. Combined with the original rear-left, rear, and rear-right, this yields a 6-camera setup that matches the input format for occupancy prediction models. Next, for each of the 6 camera images, we compute the lidar2img transformation matrix using the intrinsics and extrinsics. The images and transformation matrices are then fed into the SurroundOcc(Wei et al., 2023) model for occupancy prediction. The model outputs a 200×200×16 occupancy grid, which we use as occupancy annotations in our experiments.

Table 8: Comparison of model performance for text generation conditional and not conditional on occupancy tokens.

| Text Access to Occupancy | Speed (%) | Trajectory (%) | Overall. (%) | Inference Time (s/it) |
|:---:|:---:|:---:|:---:|:---:|
| ✓ (Setting 1) | 0.5369 | 0.7706 | 0.4245 | 21.94 |
| ✗ (Setting 2) | 0.5483 | 0.7724 | 0.4308 | 17.95 |

Table 9: Ablation study of the occupancy supervision. The ✗ indicates that the model corresponds to the original VLM without occupancy integration, whereas the ✓ denotes that the model is trained through joint occupancy–vision–language learning.

| Method | Occupancy Supervision | Speed (%) | Trajectory (%) | Avg. (%) |
|:---|:---:|:---:|:---:|:---:|
| OccVLA | ✗ | 48.55 | 55.47 | 52.01 |
| OccVLA | ✓ | 52.43 | 58.18 | 55.31 |

**Dataset Split:** Based on the distribution of meta actions, we selected 40k samples for training and generated the corresponding fine-grained annotations as described above. Experimental Results Due to limited computational resources and time constraints, we have not yet conducted full trajectory planning experiments on the test set. Instead, we evaluate our model on the validation set of WOD-E2E to assess the meta-action prediction accuracy. Table below summarizes the results with and without occupancy supervision. As shown in Table 9, Occupancy supervision leads to an overall improvement of 3.3%, which is significantly higher than the 1.5% gain reported on nuScenes. This indicates that on the more challenging long-tail WOD-E2E dataset, our joint occupancy supervision provides stronger benefits and enhances the model's perception robustness in rare or complex scenarios.

Since meta actions provide high-level guidance for downstream trajectory planners, we believe that the observed improvements in meta action accuracy validate the effectiveness and generalization of OccVLA. Finally, given that the WOD-E2E dataset lacks LiDAR data and thus provides only limited-quality occupancy labels, we expect that OccVLA could achieve even stronger performance under more complete and accurate occupancy supervision.

## D  PROMPT AND ANNOTATION DETAILS

We present a complete example of how we generate a single datapoint in our dataset, OccVLA-nuScenes. We select one frame from a driving scene of nuScenes and show the three separate conversations with GPT-4o for generating *meta action*, *scene description*, and *reasoning*.

### D.1  SCENE INFORMATION

- **Scene ID:** 0002
- **Frame Token:** 244e2876a6744249814936b24b6b03ea
- **Location:** Singapore One North

### D.2  META ACTION ANNOTATION

The meta action annotations comprise two aspects: speed and trajectory.

For the **speed meta action**, we determine the category by calculating the ego-vehicle's acceleration ($a$) at the current timestep. The action is classified as *Accelerate* if $a > 1$ km/h/s, *Decelerate* if $a < -1$ km/h/s, and *Maintain Speed* otherwise.

For the **trajectory meta action**, annotations are generated using GPT-4o. Table 10 presents the prompt employed to generate trajectory meta action annotations for the OccVLA-nuScenes dataset.

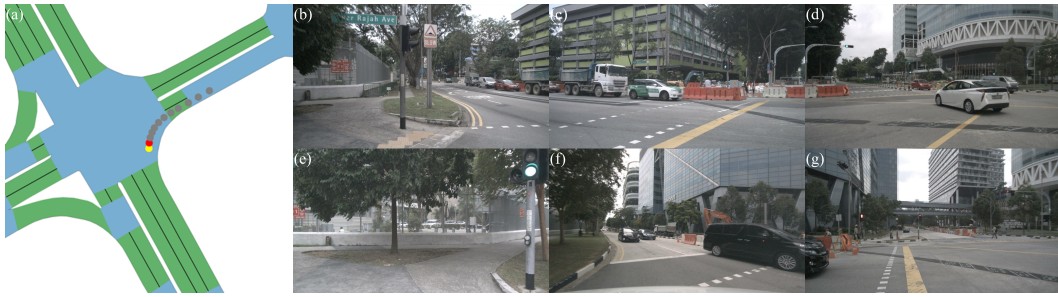

Figure 7: Illustration of the selected frame used in the example datapoint. (a) Bird's-eye view of the driving map, where the yellow dot marks the current ego-vehicle position, the red dot marks the position in the next frame, and the grey dots indicate future positions along the planned trajectory. (b)–(g) Images captured from the six on-board cameras: (b) front-left, (c) front, (d) front-right, (e) rear-left, (f) rear, and (g) rear-right views.

---

**System Prompt:**
You are an advanced AI assistant specializing in analyzing autonomous driving data.

---

**Input Prompt:**
Your task is to categorize the motion of a car into one of the following actions at the current timestamp:
- Go Straight
- Turn Left
- Turn Right
- Bear Left
- Bear Right
- Stop

You will be given the following inputs:
Map: A map showing the car's position at the current timestamp (marked as a yellow dot), its position 0.5 seconds later (marked as a red dot) and its future positions (marked as gray dots). The map also includes the road layout, lanes, and intersections. The map is always oriented with the car's current direction at top.
Current camera image: A front-facing image from the car's camera at the current timestamp.
Next camera image: A front-facing image from the car's camera 0.5 seconds after the current timestamp.

Your analysis should follow these steps:
Analyze the map: Determine how the car's position (yellow dot to red dot) has changed relative to the road layout, lanes, and intersections. Focus specifically on the immediate trajectory, leveraging the gray future trajectory points only as a supplementary reference for the current decision. Be cautious not to incorporate data from far future points that do not impact the present driving action.
Analyze the camera images: Look for visual cues in the current and next camera images, such as lane markings, road curvature, intersection geometry, or stopped vehicles, to refine your understanding of the car's motion.
Combine findings: Use the information from the map and camera images to reason about the car's trajectory and classify its motion into one of the predefined categories.

Output the result: Clearly state the category and provide a brief explanation for your reasoning. Your output should be directly json format, without any markdown syntax or additional text.

---

**Response:**
{
    "Map Analysis": "The map shows the car's current location as a yellow dot and its next location as a red dot. The trajectory of gray dots indicates a gradual curve to the right, suggesting that the car is not moving straight but is instead following a rightward path. The road layout confirms this as the car is navigating through an intersection and aligning with a right-turn lane.",
    "Camera Image Analysis": "The current camera image shows the car approaching an intersection with barriers and a clear right-turn lane. The next camera image confirms that the car has moved further into the right-turn lane, with the barriers and intersection geometry reinforcing the rightward motion.",
    "Combined Findings": "Both the map data and the camera images confirm that the car is executing a right turn at the intersection. The trajectory and visual cues align with this classification.",
    "Meta-Action": "Turn Right"
}

---

Table 10: Example of text prompts used for generating trajectory meta actions.

### D.3 Chain-of-Thought (CoT) Annotation with GPT-4o

To generate our Chain-of-Thought annotations, we employ a multi-stage process. We first prompt GPT-4o to produce a detailed **scene description**, as shown in Table 11.

Subsequently, this description is used as context to generate the **CoT reasoning traces**, illustrated in Table 12.

These traces provide step-by-step explanations of the decision-making process for motion planning tasks.

---

**System Prompt:**
You are a autonomous driving labeler. You have access to two panoramic images that encapsulates the surroundings of a vehicle in a 360-degree view. Each panoramic image is a composite of three smaller images. The first image depicts scenes to the left-front, directly in front, and right-front of the vehicle. The second image, displays views of the left-rear, directly behind, and right-rear of the same vehicle. As a driver on the road, you should follow any common sense traffic rules. You should try to stay in the middle of your lane. You should maintain necessary distance from the leading vehicle. You should observe lane markings and follow them. Your task is to do your best to understand the driving scene. If you cannot provide a response people will get injured.

---

**Input Prompt:**
Given two panoramic images, your task is to analyze and interpret the current driving scene. You should summarize the driving scenario in a paragraph. Provide a detailed but concise description of the driving scene. For example, specify the road condition, noting any particular settings (parking lot, intersection, roundabout), any important traffic elements (pedestrain, vehicle, traffic sign/light), time of the day and weather.

Be aware of the following:
- When describing the traffic elements, please specify their location or appearance characteristics to make them more distinguishable.
- Do not mention the "first/second image", ""front-left/rear-center view", etc. Instead, replace it with what is present at specific vehicle positions (front, back, left, right, etc.).
- Always answer as if you are directly in the driving scene.
- Do not merely mention generic traffic rules; integrate the information from the image. For example, if there exists a traffic light in the image, you should mention its color and influence on the driving scene.
- Answer based only on the content determined in the image, and do not speculate on uncertain content.

---

**Response:**
Cloudy daytime: At an urban intersection, a green light signals vehicles to proceed. To the front-left, a road sign marks 'Ayer Rajah Ave,' with a 'SLOW' warning sign nearby, while a fenced sidewalk runs adjacent to the road. Directly ahead, a white dump truck and a green taxi wait at the junction, bordered by red barricades, construction equipment, and a green building. To the front-right, a modern glass building and a white sedan entering the intersection are visible. Behind, a black van and a few other vehicles travel along a lane flanked by construction barriers and modern office buildings. Pedestrians are sparse, and the scene is accompanied by ongoing construction activity.

---

Table 11: Example of text prompts used for generating scene descriptions.

# E    TRAJECTORY PLANNING TASK: OCCUPANCY VS. NON-OCCUPANCY COMPARISON

As shown in Fig. 8 and Fig. 9, we present qualitative examples of trajectory planning task responses under two settings: training with occupancy information and without occupancy information. In the scene descriptions, the model primarily focuses on nearby critical objects that may impact driving, providing a more comprehensive and accurate understanding compared to the setting without occupancy. Meanwhile, the occupancy predictions further offer interpretability for the model's perception process.

# F    EXPERIMENTAL SETTINGS AND HYPERPARAMETERS

We report the detailed experimental configurations, including training parameters, model architectures, and dataset splits.

# G    LLM USAGE STATEMENT

In the preparation of this manuscript, we used GPT-5 solely for language polishing purposes. The model was instructed to refine the grammar, clarity, and readability of the text without altering the original meaning of the content.

All outputs generated by the LLM were carefully reviewed and verified by the authors to ensure that no false, misleading, or controversial information was introduced. The polished language faithfully conveys the authors' intended meaning.

**System Prompt:** You are an expert autonomous driving analyst. As a driver on the road, you should follow any common sense traffic rules. You should try to stay in the middle of your lane. You should maintain necessary distance from the leading vehicle. You should observe lane markings and follow them. Your task is to do your best to understand the driving scene. If you cannot provide a response people will get injured.

**Input Prompt:** Given the current scene and related information, explain the reasoning process that would lead to the given future meta-actions. Each meta-action represents the vehicle's state at 0.5-second intervals.

Input format:
1. Scene images: [Front, Back views of current moment]
2. Scene description: [Detailed description of important objects in the current traffic scenario]
3. Past meta-actions (t-5 to t-1): [Velocity and trajectory changes for past 2.5 seconds, at 0.5s intervals]
4. Future trajectory meta-actions (t+0 to t+6): [Known future speed and trajectory changes for next 3 seconds, at 0.5s intervals]

Task:
Provide a detailed step-by-step reasoning process that would logically lead to the given future meta-actions. Since the scene description already covers the objects present, focus on analyzing their impact on driving decisions.

Structure your analysis as follows:

1. Future meta-action analysis:
- What driving decisions (trajectory and speed) can be inferred from the future meta-actions?
- *Keep your reasoning in one sentence*

2. Situation Assessment:
- How do the described objects influence potential driving decisions? These influences should match the future meta-actions. But you should not directly use or mention the future meta-actions in this section.
- What are the immediate safety considerations?
- *Keep your reasoning in two or three sentences*

3. Historical Pattern Analysis:
- **In this section, you should not use the future meta-actions**
- What patterns emerge from the past 2.5 seconds of meta-actions?
- What driving strategy has been demonstrated so far?
- *Keep your reasoning in two or three sentences*

4. Forward Reasoning:
- **In this section, you should explain your reasoning as if you don't know the future meta-actions, but show how logical deduction would lead to these actions.**
For each 0.5s timestep (t+0 to t+6):
- What factors would drive the decision at this moment?
- Why would this specific meta-action be the optimal choice?
- *Keep your reasoning in one sentence*

Remember: for all your output, do not use ambiguous terms like "adjust trajectory and speed" or "change speed." Instead, use specific terms like "accelerate," "decelerate," "maintain speed," "turn left," "turn right," etc.

**Response:**
**Future meta-action analysis:** The future meta-actions indicate that the vehicle will execute a sustained left turn while initially slowing down, then maintaining speed through the turn, and finally accelerating upon exiting the turn.
**Situation Assessment:** Cloudy afternoon: The scene contains construction barriers, traffic lights, and vehicles positioned at an intersection, suggesting a complex environment requiring careful navigation. Immediate safety considerations include avoiding collisions with nearby vehicles and maintaining control while negotiating the turn amidst potential obstacles and construction zones.
**Historical Pattern Analysis:** Over the past 2.5 seconds, the vehicle has consistently executed left turns while maintaining or increasing speed, demonstrating a strategy of navigating a leftward trajectory with stable control. The acceleration in the last timestep suggests preparation for a more dynamic maneuver or clearing the intersection efficiently.
**Forward Reasoning:**
t+0 (0s): While turning left around the corner, the vehicle should decelerate to safely follow the bend and account for the visible barriers ahead.
t+1 (0.5s): Continuing the left turn, the car should hold its current speed to smoothly align with the road's curve and avoid the construction zone.
t+2 (1.0s): The leftward trajectory should be maintained at a steady pace, ensuring the vehicle stays clear of surrounding traffic in the intersection.
t+3 (1.5s): As the turn progresses, keeping a constant speed will help the driver maintain control and navigate past nearby obstacles.
t+4 (2.0s): The vehicle can start accelerating slightly while still turning left, using the added speed to transition onto the next road segment.
t+5 (2.5s): Increasing speed while completing the left turn will aid in merging smoothly into the lane beyond the intersection.
t+6 (3.0s): With the turn nearly finished, the vehicle should continue accelerating to match the traffic flow and adapt to upcoming road conditions.

Table 12: Example of text prompts used for generating annotations of the Chain-of-Thought (CoT) reasoning process.

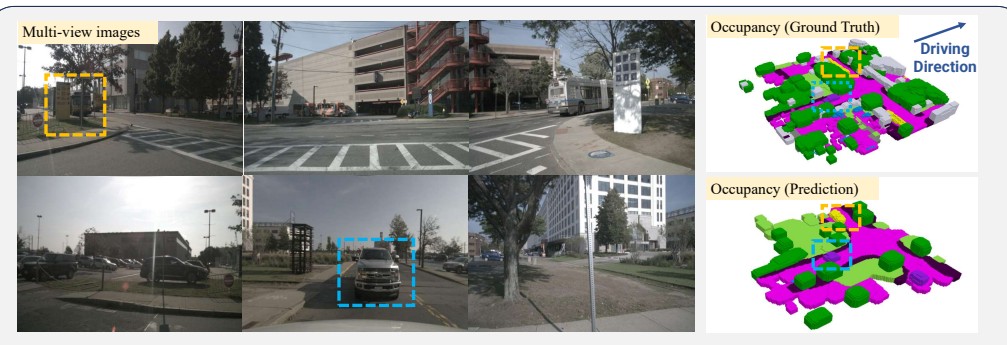

**Training w/ Occupancy:**
[(Maintain speed, Go Straight), (Maintain speed, Go Straight),
(Maintain speed, Go Straight), (Maintain speed, Go Straight),
(Maintain speed, Go Straight), (Maintain speed, Go Straight),
(Maintain speed, Go Straight)],

**Description:**
Sunny afternoon: The vehicle is at an intersection with a zebra crossing directly ahead. To the left-front, a white truck is parked near a sidewalk lined with trees and a sidewalk. Directly in front, a pedestrian crossing sign is visible near the crosswalk. To the right-front, a pedestrian crossing sign is visible near a crosswalk. Behind the vehicle, a white delivery truck is parked on the left side of the road, while a pedestrian crossing sign is visible on the sidewalk near the intersection. To the right-rear, a pedestrian crossing sign is visible near the corner of a modern building. The road markings are clear, and the weather is bright and sunny, providing good visibility.

**Intention:**
Over the past 2.5 seconds, the vehicle has transitioned from a complete stop, indicating a cautious approach to the intersection. The consistent trajectory suggests the vehicle is focused on maintaining lane discipline and preparing for a clear path ahead. The vehicle is stopped at a red light at the intersection, requiring the vehicle to remain stationary until it turns green. The pedestrian crossing to the right-front poses a potential hazard if they veer into the crosswalk, while the clear road ahead allows for continued forward movement.

**Training w/o Occupancy:**
[(Accelerate, Go Straight), (Accelerate, Go Straight),
(Accelerate, Go Straight), (Accelerate, Go Straight),
(Accelerate, Go Straight), (Accelerate, Go Straight),
(Accelerate, Go Straight)]

**Description:**
Sunny afternoon: The vehicle is at an intersection with clear lane markings. To the front-left, a sidewalk lined with trees and a 'Do Not Enter' sign is visible. Directly ahead, the road is clear, flanked by trees and buildings. To the front-right, a pedestrian crossing sign is visible near a crosswalk. Behind the vehicle, a white truck is parked on the left side of the road near a bus stop on the right. To the right-rear, a parking lot with several parked cars is visible. The surroundings include a mix of industrial and commercial buildings, with clear skies enhancing visibility.

**Intention:**
Over the past 2.5 seconds, the vehicle has transitioned from maintaining speed to accelerating, indicating a cautious approach to the intersection and readiness to proceed. The consistent "Go Straight" trajectory suggests the vehicle is preparing to proceed once the path is clear. The pedestrian crossing ahead requires the vehicle to remain stopped until the crossing is clear, while the pedestrian crossing to the left poses no immediate threat as they are on the sidewalk. The clear weather and good visibility reduce the likelihood of unexpected hazards, but the presence of pedestrians near the crosswalk.

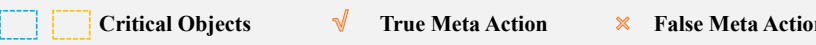

Figure 8: Visualization for trajectory planning tasks.

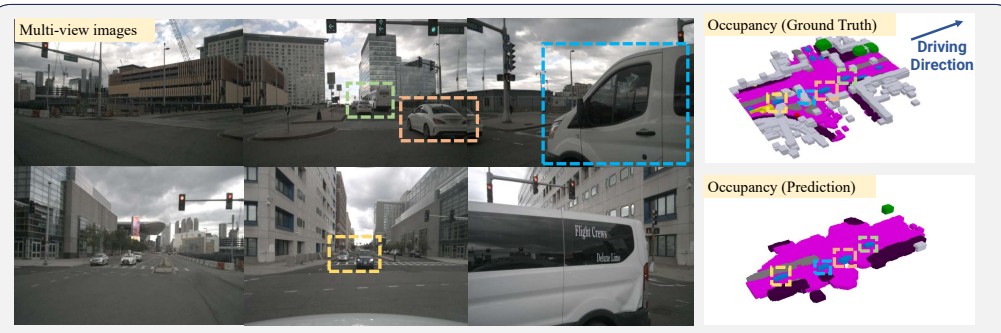

**Training w/ Occupancy:**

[(Maintain speed, Go Straight), (Maintain speed, Go Straight),
(Maintain speed, Go Straight), (Maintain speed, Go Straight),
(Maintain speed, Go Straight), (Maintain speed, Go Straight),
(Maintain speed, Go Straight)],

**Description:**

Cloudy afternoon: The vehicle is at a city intersection surrounded by modern high-rise buildings. Ahead, a green traffic light signals the right of way, while the traffic light is green, allowing vehicles to proceed. To the left-front, a white SUV is parked near a sidewalk lined with trees and a parking structure. To the right-front, a construction site with cranes and partially built structures dominates the view. Behind, a line of vehicles, including a black SUV and a silver sedan, waits at the light. The road is wet, suggesting recent or ongoing rain, and visibility is slightly reduced due to the rain.

**Intention:**

Over the past 2.5 seconds, the vehicle has consistently accelerated while maintaining a straight trajectory, indicating a focus on progressing through the intersection efficiently. The lack of lateral adjustments suggests no immediate need for lane changes or evasive maneuvers. The green traffic light ahead allows the vehicle to proceed without stopping, while the wet road conditions necessitate cautious driving to maintain traction and avoid skidding. The green traffic light ahead allows for continued forward movement, but the presence of the white SUV in the adjacent lane requires maintaining a safe following distance.

**Training w/o Occupancy:**

[(Accelerate, Go Straight), (Accelerate, Go Straight),
(Accelerate, Go Straight), (Accelerate, Go Straight),
(Accelerate, Go Straight), (Accelerate, Go Straight),
(Accelerate, Go Straight)]

**Description:**

Cloudy afternoon: At an urban intersection, the traffic light ahead is green, allowing vehicles to proceed. To the left-front, a white pickup truck is parked near a sidewalk bordered by modern office buildings. Directly in front, a black sedan waits at the intersection, while a black sedan waits at the intersection. To the right-front, a pedestrian walks along the sidewalk near a 'No Turn on Red' sign. Behind the vehicle, a line of vehicles, including a black Jeep and a black sedan, follow closely in adjacent lanes. To the left-rear, a white pickup truck is parked near the curb. The road is clear of pedestrians, and the weather is overcast with light rain.

**Intention:**

Over the past 2.5 seconds, the vehicle has consistently accelerated while maintaining a straight trajectory, indicating a focus on progressing through the intersection efficiently. The lack of lateral adjustments suggests no immediate need for evasive maneuvers or lane changes. The green traffic light ahead allows the vehicle to proceed, while the pedestrian crossing on the right requires vigilance to ensure they clear the crosswalk. The closely following black sedan behind necessitates smooth and predictable driving to avoid abrupt braking that could lead to a rear-end collision.

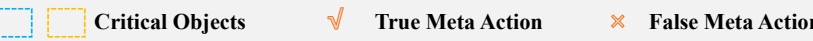

⬜ 🟧 **Critical Objects**          √ **True Meta Action**          ✗ **False Meta Action**

Figure 9: Visualization for trajectory planning tasks.

| Configuration | Motion Planning | | | VQA |
| --- | --- | --- | --- | --- |
| | Stage 1 | Stage 2 | Stage 3 | nuScenes-QA |
| Training component | Full | Adapter | Planning Head | Adapter |
| Learning rate | 3E-5 | 3E-5 | 1E-4 | 2E-5 |
| Warmup step | 500 | 500 | 0 | 500 |
| Weight decay | 1E-6 | 1E-6 | 1E-6 | 1E-6 |
| Batch size | 64 | 64 | 64 | 64 |
| $\lambda_{occ}$ | - | 1 | - | 1 |
| Optimizer | AdamW (Loshchilov & Hutter, 2017) | | | |
| Dataset | OmniDrive | OccVLA-nuScenes | | nuScenes-QA |
| Training epoch | 2 | 5 | 20 | 3 |

Table 13: Experimental settings and hyperparameters.

