# OpenReview forum: "OccVLA: Vision-Language-Action Model with Implicit 3D Occupancy Supervision"
_ICLR.cc/2026/Conference — Submitted to ICLR 2026_

### Official Review · Reviewer_ho2c · 2025-10-23

**Soundness:** 3
**Presentation:** 2
**Contribution:** 3
**Rating:** 4
**Confidence:** 4

**Summary:**

This paper presents a novel framework called OccVLA, which integrates 3D occupancy supervision into a VLA model. The model is trained to predict latent occupancy representations, while the occupancy reasoning can be skipped during inference. Experimental results demonstrate that OccVLA achieves strong performance on the nuScenes planning and QA tasks.

**Strengths:**

1.	OccVLA presents an effective approach to integrating occupancy supervision and demonstrates promising results on the nuScenes planning and QA tasks.

2.	The method introduces no additional inference cost.

3.	The visualization in Figure 5 shows that OccVLA can generate meaningful occupancy representations.

**Weaknesses:**

1.	In OccVLA training, both occupancy supervision and CoT supervision are used jointly. It would be clearer to isolate the contribution of each component and provide detailed ablation studies. Additionally, it is unclear which model serves as the baseline that excludes only the OccVLA-nuScenes training.

2.	It remains unclear how occupancy prediction is removed during inference and how much the resulting misalignment between training and testing affects performance. Furthermore, is this skipping process a default setting used in all experiments or an optional acceleration setting?

3.	(Minor) It looks like the authors modify the line space significantly. The captions of Figure 1 and Figure 5 are nearly overlap with the following text.

4.	(Minor) There is a spelling error in Line 299, where the word “choozaizuose” is misspelled.

**Questions:**

See weaknesses.

---

> ### Author Response · Authors · 2025-11-22
> **Clarifications and Additional Analyses on OccVLA: Component Contributions and Inference Efficiency**
>
> We sincerely thank all reviewers for their constructive feedback and valuable suggestions. Below, we address each point in detail and provide additional analyses and experiments to further strengthen our paper.
> ## 1. Component Contributions and Ablation Study
> We sincerely thank the reviewers for their valuable comments regarding the joint use of occupancy supervision and chain-of-thought (CoT) supervision during OccVLA training. To better illustrate the independent contributions of each component, we conducted a detailed ablation study by selectively removing different supervision signals and analyzing the corresponding performance changes. The results are shown below:
>
> | CoT | Occupancy | Speed (%) | Trajectory (%)| Overall. (%)|
> | --- | --------- | -------- | --------- | ----- |
> | ×   | ×            | 53.05   | 76.25   | 40.26 |
> | ✓   | ×           | 53.77   | 77.24    | 41.48 |
> | ×   | ✓             | -      | -       | -     |
> | ✓   | ✓            | 54.83   | 77.95   | 43.08 |
> As shown in the table, removing both CoT and occupancy supervision leads to a significant performance drop. Introducing CoT supervision alone improves trajectory and speed prediction accuracy, indicating that structured reasoning benefits planning tasks. When CoT and occupancy supervision are jointly applied, the model achieves the best overall performance. This suggests that occupancy supervision enhances spatial understanding of driving scenes while complementing CoT-based reasoning.
> We also tested a setting where only occupancy supervision was used without CoT. This configuration led to overfitting to the occupancy signal, causing repetitive outputs during autoregressive generation. The baseline model here corresponds to training without the OccVLA-nuScenes dataset, i.e., without occupancy or CoT supervision, relying solely on meta-action training of the VLM model.
> ## 2. Removing Occupancy Prediction During Inference
> We appreciate the reviewers’ questions regarding how occupancy prediction is removed during inference and whether this introduces a mismatch between training and testing. To verify this, we trained and tested the model performance under two configurations:
> - Setting 1: Text generation with occupancy tokens
>   Text generation is conditioned on both image tokens and occupancy tokens. The model explicitly predicts occupancy maps and incorporates them into the reasoning process.
> $$P(Y|I, O) = \prod_{t=1}^{T} P(y_t | y_{<t}, I, O) $$
> - Setting 2: Text generation without occupancy tokens
>   Text generation is conditioned only on image tokens. During training, occupancy supervision is still used to update model parameters via cross-attention with image tokens. However, at inference time, text generation does not access occupancy tokens, effectively skipping occupancy prediction.
> $$P(Y|I) = \prod_{t=1}^{T} P(y_t | y_{<t}, I) $$
> We trained and evaluated the model under both configurations, and the results are summarized below:
>
> | Text Access to Occupancy  | Speed (%) | Trajectory (%)   | Overall. (%) | Inference Time (s/it) |
> | ------------------------ | -------- | -------- | --------- | --------------------- |
> | ✓ (Setting 1)                         | 53.69  | 77.06 | 42.45    | 21.94                 |
> | ×  (Setting 2)                       | 54.83  | 77.24    | 43.08    | 17.95                 |
>
>  Setting 2 provides approximately a 25% speed improvement. Interestingly, it also yields a slight performance gain of about 0.5%. We hypothesize that excessive visual tokens (occupancy tokens) may introduce redundant information that interferes with reasoning, leading to minor degradation. This result demonstrates that our approach achieves both efficiency and accuracy advantages during inference.
>
> We report the experimental results based on **Setting 2 in this paper**, under which OccVLA supports flexible inference modes: users can choose to skip occupancy prediction for faster inference or retain it to visualize the model’s spatial understanding of 3D scenes. Overall, this skipping mechanism serves as an optional acceleration setting rather than a default configuration. It does not introduce a significant mismatch between training and inference stages. The model exhibits robust and consistent performance across both modes.
> ## 3. (Minor) Line spacing issue
> We acknowledge the reviewer’s observation regarding the line spacing. We will fix this formatting issue in the revised version of the paper.
> ## 4. (Minor) Spelling error
> We appreciate the reviewer’s careful attention. This will be corrected in the updated version of the manuscript.
>
> We hope that our clarifications and new results address all concerns. If there are any remaining questions or points requiring further explanation, we would be happy to provide additional details.

---

### Official Review · Reviewer_E7ic · 2025-10-24

**Soundness:** 3
**Presentation:** 2
**Contribution:** 3
**Rating:** 4
**Confidence:** 5

**Summary:**

This paper introduces OccVLA, a Vision-Language-Action model designed to integrate 3D spatial understanding critical for autonomous driving. It addresses the limitation of current Multimodal Large Language Models (MLLMs) by treating dense 3D occupancy as an implicit supervisory signal, allowing the model to learn fine-grained spatial structures directly from 2D input. Crucially, the 3D occupancy prediction is an intermediate reasoning step that can be skipped during inference, ensuring no added computational overhead or latency degradation. The comprehensive experiments demonstrate the model's effectiveness across various tasks, including VQA, grounded perception, and autonomous driving planning on standard benchmarks.

**Strengths:**

1. The concept of utilizing implicit 3D occupancy as a self-supervisory signal represents a significant contribution to spatial reasoning in autonomous driving.
2. The unified architecture is capable of handling diverse multimodal tasks (perception, reasoning, and action planning) within a single, consistent framework.

**Weaknesses:**

1. The core concept bears significant resemblance to the approach proposed in previous works ([*] Ross: Reconstructive Visual Instruction Tuning, ICLR2025), yet the paper fails to discuss or provide a comparative analysis against these related works.
2. The paper introduces the "Occupancy Transformer" as a core component, yet it fails to clearly articulate its novel architectural differences compared to a standard transformer block.
3. The description of the Latent Occupancy Prediction module is incomplete in several crucial aspects. Firstly, the motivation for choosing VQ-VAE over alternative modern generative models, such as diffusion-based variants, is insufficiently discussed. Secondly, many critical architectural hyperparameters are missing, most notably the specific number of queries used for the transformer-based prediction.
4. The evaluation of the planning capability is limited to open-loop metrics on the nuScenes dataset, lacking essential simulation or closed-loop experiments.
5. A significant concern remains that the overall complexity and integrated architecture—even without explicitly activating the 3D occupancy prediction—still introduces an inherent latency overhead compared to simpler, pure VLA baselines. A detailed latency profile comparing the final OccVLA model (with the module skipped) against relevant efficient baselines and against its own full model variant (with the occupancy module included) is essential for validation.
6. Line 299 contains a clear typo ("choozaizuose").

**Questions:**

None

---

> ### Author Response · Authors · 2025-11-28
>
> ## 1. Comparison with Ross (Reconstructive Visual Instruction Tuning, ICLR 2025)
> We sincerely thank the reviewer for pointing out the similarity with Ross: Reconstructive Visual Instruction Tuning (ICLR 2025) . After carefully reviewing this work, we confirm that our proposed OccVLA framework differs substantially from Ross in both principle and contribution. Ross employs reconstruction loss to supervise visual tokens generated by the LLM, focusing exclusively on 2D visual reconstruction. In contrast, OccVLA maps visual tokens into 3D occupancy representations based on cross-attention, enabling cross-modal understanding from 2D vision to 3D spatial reasoning. This key distinction reflects a fundamentally different objective, enhancing 3D spatial perception rather than reconstructing visual content. We will include a detailed discussion of Ross in the updated version of the paper’s Introduction section.
> ## 2. Architectural Novelty of the Occupancy Transformer
> While our design adopts the standard transformer framework, its novelty lies in how trainable occupancy queries are embedded directly within the transformer’s cross-attention structure. These queries interact with visual features to infer occupancy distributions, effectively integrating 3D spatial prediction into the transformer’s decoding process. This design ensures scalability and seamless compatibility with decoder-only language model architectures.
> ## 3. Design Rationale and Implementation Details of the Latent Occupancy Prediction Module
> We understand the reviewers’ concerns regarding the rationale for adopting VQ-VAE and the missing architectural details.
> First, we agree that diffusion-based generative approaches possess strong potential for fine-grained 3D generation and rendering. However, the primary objective of OccVLA is not high-fidelity 3D synthesis, but rather enhanced 3D understanding and spatial reasoning. Diffusion models typically require multiple denoising iterations per sample and extensive training epochs to converge, which is inconsistent with the relatively lightweight training paradigm of vision-language models (VLMs). Although advanced multimodal large language model (MLLM) frameworks such as Transfusion and LMfusion enable joint training of diffusion-based image generation and autoregressive text generation, such paradigms remain impractical in autonomous driving scenarios due to the limited scale of available datasets.
> In contrast, our approach leverages a pretrained VQ-VAE decoder, requiring the model to align only with the latent space of VQ-VAE. The VQ-VAE-based representation leads to more efficient training and better matches the optimization characteristics of VLMs, making it a more suitable choice for our framework.
> Specifically, our Transformer-based occupancy prediction module employs N = 2500 learnable queries, following the design of OccWorld  (ECCV 2024, “Reconstructive Visual Instruction Tuning”).
> ## 4. Evaluation on Planning Capability and Dataset Choice
> We appreciate the reviewer’s concern about the limited scope of our evaluation. The nuScenes dataset is one of the few autonomous driving benchmarks that simultaneously provides trajectory planning annotations and rich perception tasks (e.g., occupancy prediction and VQA). Unfortunately, most existing simulation or closed-loop datasets do not include occupancy ground truth, which constrains our evaluation to nuScenes. Furthermore, prior works such as OccWorld (ECCV 2024, “Reconstructive Visual Instruction Tuning”), OccLLaMA (arXiv:2409.03272, “A Unified Occupancy-Language-Action World Model for Understanding and Generation Tasks in Autonomous Driving”), and Occ-LLM (arXiv:2502.06419, “Enhancing Autonomous Driving with Occupancy-Based Large Language Models”) have also primarily evaluated VQA and planning capabilities on the nuScenes dataset, which has become a widely adopted benchmark in this line of research. Within this established context, our experiments are designed to align with prevailing evaluation practices in the field, ensuring comparability while providing a comprehensive assessment under the open-loop setting.

---

> ### Author Response · Authors · 2025-11-28
>
> ## 5. Latency and Efficiency Analysis
> We thank the reviewer for raising the important question regarding latency and inference efficiency. To address this, we conducted experiments under two configurations:
> - Setting 1: Text generation with occupancy tokens
> Text generation is conditioned on both image tokens and occupancy tokens, where the model explicitly predicts occupancy maps and integrates them into reasoning:
> $$P(Y|I, O) = \prod_{t=1}^{T} P(y_t | y_{<t}, I, O) $$
> - Setting 2: Text generation without occupancy tokens
> Text generation is conditioned only on image tokens. During training, occupancy supervision is still applied via cross-attention, but at inference time, the model skips occupancy prediction:
> $$P(Y|I) = \prod_{t=1}^{T} P(y_t | y_{<t}, I) $$
> The comparative results are summarized below:
> | Text Access to Occupancy  | Speed (%) | Trajectory (%)   | Overall. (%) | Inference Time (s/it) |
> | ------------------------ | -------- | -------- | --------- | --------------------- |
> | ✓ (Setting 1)                         | 53.69  | 77.06 | 42.45    | 21.94                 |
> | ×  (Setting 2)                       | 54.83  | 77.24    | 43.08    | 17.95                 |
>
>
>
> Setting 2 achieves approximately 25% faster inference and even shows a slight performance improvement (~0.5%). We hypothesize that excessive visual tokens (i.e., occupancy tokens) may introduce redundant information that mildly interferes with reasoning. These findings demonstrate that OccVLA achieves both efficiency and accuracy advantages during inference.
> The experimental results reported in this paper are based on Setting 2, where the text generation process does not depend on occupancy-conditioned inputs. Consequently, the occupancy prediction step can be safely skipped without affecting overall performance. On this basis, we emphasize that OccVLA supports a flexible inference paradigm: users may choose to omit occupancy prediction for faster inference or retain it to visualize the model’s spatial understanding of 3D scenes.
> ## 6. Typographical Error
> We appreciate the reviewer’s attention to detail. This will be corrected in the final version.

---

### Official Review · Reviewer_QBwy · 2025-10-30

**Soundness:** 1
**Presentation:** 3
**Contribution:** 2
**Rating:** 4
**Confidence:** 5

**Summary:**

The paper proposes OccVLA, a novel Occupancy Vision-Language-Action framework that integrates dense 3D occupancy prediction into the VLM backbone, enabling the model to learn fine-grained spatial understanding directly from 2D images. This approach treats occupancy prediction as an implicit reasoning process that can be skipped during inference for zero computational overhead, achieving state-of-the-art results on nuScenes for trajectory planning and 3D visual question-answering.

**Strengths:**

1. The paper is highly motivated. Using a dense 3D signal to help the VLM/MLLM improve the spatial understanding capability is a novel idea.
2. The paper is easy to follow.
3. The methods demonstrate SOTA performance on two benchmarks.

**Weaknesses:**

I have major concerns in the paper's incorrect statements and experiments.
1) The authors make an invalid claim regarding existing VLM supervision. The statement in the introduction section that "supervision relies on 3D annotations described in text (e.g., coordinates or bounding boxes), which are inherently weak and sparse"  lacks supporting evidence. While bounding boxes are inherently sparser than dense occupancy, the term 'weak' is judgmental and should either be supported by a metric-based comparison or replaced with a more neutral term.
2) The argument against the scalability of 3D bounding box supervision is weakened by recent advances in auto-labeling. The critique that prior methods are constrained by a lack of scalability due to the need for "extensive manual labeling" is outdated. Given that the proposed OccVLA-nuScenes dataset relies on automated pipelines to generate occupancy, the authors should acknowledge that recent advancements have also enabled high-quality auto-labeling for 3D detection/bounding box supervision, thereby allowing both dense (occupancy) and sparse (box) supervision methods to scale.
3) The characterization of the EMMA baseline is inaccurate and potentially misleading. The paper claims that the state-of-the-art method EMMA relies on costly supervision (3D/BEV coordinates & 3D bounding box) that "limits its scalability". This is incorrect for the base model, which achieves strong results via self-supervision on motion planning trajectories alone. The authors should clarify the specific EMMA variant they are comparing against and justify why its data requirements fundamentally limit scalability more than their own approach, which requires dense, automatically generated 3D occupancy data.
4) The related work section and experimental evaluation lack important context. (1) The authors should discuss recent and relevant work such as S4-Driver[1], which also focuses on improving Large Language Models for driving by enhancing spatio-temporal visual representation and lifting 2D information to 3D. (2) The nuScenes planning benchmark is rapidly becoming outdated for challenging scenarios. To demonstrate the robust performance of OccVLA on the latest benchmarks and its capability in long-tail scenarios, the authors are strongly encouraged to provide results on the current standard, such as the Waymo Open Dataset End-to-End Driving (WOD-E2E) challenge[2].

[1]S4-Driver: Scalable Self-Supervised Driving Multimodal Large Language Modelwith Spatio-Temporal Visual Representation
[2] WOD-E2E: Waymo Open Dataset for End-to-End Driving in Challenging Long-tail Scenarios

**Questions:**

**Verification of Foundational Claims**
The paper's narrative regarding the limitations of existing supervision requires justification, especially concerning scalability and the quality of ground truth.
- Scalability of Bounding Box Supervision: The claim that models requiring 3D Bounding Boxes are not scalable due to human effort is debatable, given recent advances in high-quality auto-labeling techniques for 3D detection. To validate the paper's core motivation, the authors must address the reliance on automated pipelines for both dense and sparse supervision. I request that the authors either:
1)  Provide quantitative evidence that the auto-labeling quality of the occupancy grid map is significantly superior to modern auto-labeled 3D detection bounding boxes, or that occupancy GT requires no auto-labeling at all.
2) Acknowledge the scalability argument and adjust the paper's storyline accordingly.
- Unsupported Claim on Supervision Quality.  The paper states that 3D annotations described in text are "inherently weak and sparse." While they are sparser than dense occupancy, the term "weak" lacks a clear technical definition or reference. I request that the authors either provide evidence to justify the use of "weak"

**Comprehensive Evaluation**
- The analysis of related work is incomplete. I believe a crucial comparison with S4-Driver is necessary.
- Outdated Benchmark : The nuScenes planning benchmark is becoming outdated. To demonstrate the method's real-world generalization to challenging long-tail scenarios, I strongly recommend evaluating performance on  Waymo Open Dataset End-to-End Driving (WOD-E2E) challenge. If the method cannot produce good occupancy grid maps or competitive planning results in WOD-E2E, this indicates a severe limitation in its generalization or scalability that must be clearly discussed as a weakness.

---

> ### Author Response · Authors · 2025-11-28
>
> ## 1. On the statement regarding existing VLM supervision
> Response:
> We appreciate the reviewer’s careful observation. We acknowledge that the term “weak” may not be technically precise. In the revised manuscript, we will replace it with a more neutral and accurate expression, such as *“sparser”*, to better describe the relative density of supervision signals without implying subjective judgment.
>
> ---
> ## 2. On the scalability of 3D bounding box supervision
> Response:
> We thank the reviewer for this insightful comment. We agree that recent developments in 3D object detection have made high-quality, automatically labeled bounding boxes feasible at scale. However, we would like to clarify that 3D bounding boxes provide only coarse-grained geometric representations, capturing object extents but omitting detailed structural information. Furthermore, rare or long-tail object categories are often excluded, as it is impractical to exhaustively annotate all possible entities. These limitations restrict the scalability of bounding-box-based supervision in open-world settings.
> In contrast, occupancy-based supervision provides fine-grained voxel-level annotations that describe the geometry of nearly all visible entities in the scene. This dense representation allows Vision-Language Models (VLMs) to learn richer spatial semantics and achieve better generalization. We will revise the manuscript to acknowledge the scalability of both automated bounding box and occupancy pipelines, while emphasizing that the core advantage of our approach lies in its **fine-grained and open-world 3D supervision**.
>
> ---
> ## 3. On the characterization of the EMMA baseline
> Response:
> We appreciate the reviewer’s clarification. We acknowledge that EMMA’s scalability primarily stems from its self-supervision on motion planning trajectories, as described in its original paper. However, to enable Vision-Language Models to better understand the 3D world, EMMA also incorporates explicit 3D supervision signals. Specifically, Sections 2.2 and 3.4 of the EMMA paper mention the use of 3D bounding boxes and 3D/BEV coordinates as auxiliary supervision for scene understanding.
> We agree that our previous statement suggesting that EMMA’s supervision “limits its scalability” was imprecise. Compared with EMMA, our occupancy-based supervision provides more fine-grained 3D labels for all visible entities, offering both detailed spatial understanding and strong scalability through mature auto-labeling pipelines.
>
> ---
> ## 4. On related work and experimental evaluation
> Response:
> We thank the reviewer for highlighting these important points.
> (1) Regarding **S4-Driver**, since the method has not been open-sourced, we were unable to reproduce its results directly. However, we compared our performance against the metrics reported in their paper on the nuScenes dataset.
> (2) Regarding the Waymo Open Dataset End-to-End Driving (WOD-E2E) benchmark, we fully agree that it represents a more challenging and up-to-date evaluation standard. We are currently extending our experiments to this dataset.

---

> ### Author Response · Authors · 2025-11-28
>
> # Response to Questions
> ## Verification of Foundational Claims
> ---
> - (1) Scalability of Bounding Box Supervision
> We sincerely thank the reviewer for this constructive comment. We acknowledge that recent advances in 3D detection have significantly improved the quality and scalability of automatically generated 3D bounding boxes. However, we would like to clarify that the ground-truth (GT) generation pipeline for occupancy supervision also relies on a well-established, fully automated process that integrates multiple sensing modalities (LiDAR, camera, and radar) and has been widely validated in the literature.
> Specifically, as detailed in Occ3D (arXiv:2304.14365, *“Occ3D: A Large-Scale 3D Occupancy Prediction Benchmark for Autonomous Driving”*), the occupancy GT is generated through a multi-frame point cloud aggregation and segmentation process, followed by projection, occlusion reasoning, clustering, and voxel filling. The process is further refined via model-based and rule-based adjustments using image-view consistency checks. This multi-modal and multi-frame fusion pipeline produces fine-grained voxel-level annotations that capture both static and dynamic scene elements. Compared to auto-labeled 3D bounding boxes, occupancy GT provides dense geometric supervision for nearly all visible entities in the open world, rather than being limited to pre-defined object categories.
> Therefore, while both occupancy and bounding box supervision benefit from automated pipelines, occupancy offers a more comprehensive and fine-grained representation of the 3D environment, which is particularly beneficial for large-scale and open-world scenarios.
> Nevertheless, we acknowledge the reviewer’s point regarding scalability and will revise the paper’s narrative to reflect that both dense (occupancy) and sparse (bounding box) supervision can be achieved through automated pipelines. We will emphasize that our main advantage lies in the finer granularity and open-world coverage provided by occupancy-based supervision, rather than solely in the scalability argument.
>
> ---
> - (2) Unsupported Claim on Supervision Quality
> We appreciate the reviewer’s observation. We agree that the term “weak” used in our original manuscript was not technically rigorous. In the revised version, we will replace it with a more precise description, such as *“sparser”*.
> To further clarify the distinction, our approach employs **occupancy queries spanning 2,500 tokens**, which encode detailed spatial semantics across the entire 3D scene. This dense supervision enables the model to learn fine-grained 3D reasoning beyond object-level boundaries. In contrast, 3D bounding box supervision covers only a limited set of  pre-defined objects and cannot represent the full spatial structure of the environment. Thus, while bounding box supervision is sparse by design, occupancy-based supervision provides a more continuous and comprehensive understanding of the 3D world.
>
> ---
> ## Comprehensive Evaluation
>
> ---
> - (1) Comparison with S4-Driver
> We thank the reviewer for highlighting the importance of S4-Driver as a relevant baseline. Since S4-Driver has not been open-sourced, we were unable to reproduce its method directly. However, we compared our results against the performance metrics reported in its paper on the nuScenes dataset. As shown in Table 1, our method achieves superior performance across multiple planning horizons.
> | Method     | L2 (1s) | L2 (2s) | L2 (3s) | L2 (Avg.) |
> |-------------|-------|-------|-------|---------|
> | **S4-Driver** | **0.16** | 0.34  | 0.63  | 0.38  |
> | **Ours**    | 0.18  | **0.26** | **0.40** | **0.28** |
>
>
> (*Results for S4-Driver are taken directly from its reported nuScenes evaluation.*)
> These results demonstrate that OccVLA achieves competitive or superior performance even under comparable conditions.
>
> ---
> - (2) Evaluation on Waymo Open Dataset End-to-End (WOD-E2E)
> We appreciate the reviewer’s suggestion to evaluate our method on the Waymo Open Dataset End-to-End Driving (WOD-E2E) benchmark to further assess generalization to complex and long-tail scenarios. We are actively working to extend our experiments to this dataset. Because WOD-E2E does not provide official occupancy annotations and lacks radar data, we employ the SurroundOcc model (*CVPR 2023, “SurroundOcc: Multi-Camera 3D Occupancy Prediction for Autonomous Driving”*) to generate high-quality pseudo-occupancy labels. Based on these, we apply the OccVLA methodology to produce detailed meta-action annotations and corresponding chain-of-thought (CoT) data. Due to the extensive engineering workload of the experiments, we are still working hard to complete them and will submit the results report as soon as they are finished.

---

> ### Author Response · Authors · 2025-11-28
>
> In summary, we will:
> 1. Clarify that both occupancy and bounding box supervision can rely on automated pipelines, but occupancy provides **finer-grained, open-world 3D supervision**.
> 2. Replace the term “weak” with “sparser” and provide a clearer explanation of the difference in supervision density.
> 3. Include a detailed discussion comparing our results with S4-Driver and report ongoing experiments on WOD-E2E to demonstrate the generalization capability of **OccVLA**.
> We thank the reviewer again for these valuable suggestions, which have helped us strengthen both the technical justification and the clarity of our paper.

---

> ### Author Response · Authors · 2025-12-03
> **Extended Experiments on WOD-E2E**
>
> We thank the reviewer for the insightful comment regarding the generalization ability of OccVLA and its performance in more challenging, long-tail scenarios. Following the reviewer’s suggestion, we have evaluated the generalization and scalability of OccVLA on the Waymo Open Dataset End-to-End Driving (WOD-E2E) dataset.
>
> Dataset Construction
> The WOD-E2E dataset primarily targets long-tail driving scenarios and contains approximately 400k training samples, each with 8 camera views and ego trajectories. However, it lacks point cloud data and occupancy annotations. To construct the training data required by OccVLA, we performed the following labeling steps:
>
> - Meta-action Annotation:
> Following the definition in our paper, we decompose the meta action into trajectory (Turn Left, Turn Right, Bear Left, Bear Right, Go Straight) and speed (Stop, Accelerate, Decelerate, Maintain speed). The trajectory labels are obtained via clustering, while speed labels are derived based on acceleration thresholds. The final meta action includes both the overall trajectory direction and the second-by-second speed actions over a 5-second horizon.
>
> - COT Annotation:
> After obtaining meta actions, we used our proposed method in conjunction with GPT-4o to generate COT reasoning annotations.
>
> - Occupancy Pseudo-labeling:
> Since LiDAR data is unavailable, the original automatic labeling pipeline cannot be applied. We therefore utilized the SurroundOcc model (pretrained on nuScenes) to predict occupancy labels.
> Most existing occupancy models are designed for 6‑camera datasets such as nuScenes, while WOD-E2E provides 8 cameras. To ensure compatibility, we first align and stitch five front‑facing images (side‑left, front‑left, front, front‑right, side‑right) using the provided camera intrinsics and extrinsics. We then recrop the stitched view back into three image views, new-front‑left, new-front, and new-front‑right. Combined with the original rear‑left, rear, and rear‑right, this yields a 6‑camera setup that matches the input format for occupancy prediction models.
> Next, for each of the 6 camera images, we compute the lidar2img transformation matrix using the intrinsics and extrinsics. The images and transformation matrices are then fed into the SurroundOcc model for occupancy prediction. The model outputs a 200×200×16 occupancy grid, which we use as occupancy annotations in our experiments.
> - Dataset Split:
> Based on the distribution of meta actions, we selected 40k samples for training and generated the corresponding fine-grained annotations as described above.
> Experimental Results
> Due to limited computational resources and time constraints, we have not yet conducted full trajectory planning experiments on the test set. Instead, we evaluate our model on the validation set of WOD-E2E to assess the meta-action prediction accuracy.
> Table below summarizes the results with and without occupancy supervision. Occupancy supervision leads to an **overall improvement of 3.3%**, which is significantly higher than the 1.5% gain reported on nuScenes. This indicates that on the more challenging long-tail WOD-E2E dataset, our joint occupancy supervision provides stronger benefits and enhances the model’s perception robustness in rare or complex scenarios.
> | Method| Occupancy Supervision | Trajectory (%)| Speed (%) | Avg.(%) |
> |----------------|----------------|--------------------|---------------|----------------------|
> | OccVLA         | ×              | 55.47               | 48.55          | 52.01                 |
> | OccVLA         | √              | 58.18               | 52.43          | 55.31                 |
> Since meta actions provide high-level guidance for downstream trajectory planners, we believe that the observed improvements in meta action accuracy validate the effectiveness and generalization of OccVLA.
> Finally, given that the WOD-E2E dataset lacks LiDAR data and thus provides only limited-quality occupancy labels, we expect that OccVLA could achieve even stronger performance under more complete and accurate occupancy supervision.

---

### Official Review · Reviewer_5Rfz · 2025-10-31

**Soundness:** 3
**Presentation:** 2
**Contribution:** 2
**Rating:** 6
**Confidence:** 5

**Summary:**

This paper proposes OccVLA, a framework that integrates 3D occupancy representations into vision-language models for autonomous driving. The key innovation is treating occupancy prediction as both an output and supervisory signal, which can be skipped during inference. The authors evaluate on nuScenes for trajectory planning and 3D visual question answering, achieving competitive results.

**Strengths:**

1.Using occupancy as implicit supervision rather than explicit input is creative and addresses computational efficiency concerns during inference.
2.The paper evaluates on multiple tasks (motion planning, VQA, occupancy prediction) and provides extensive comparisons with relevant baselines.
3.The automated data pipeline for generating meta-actions and CoT annotations could be valuable for the community.
4.Achieves state-of-the-art on trajectory planning (0.28m average L2 error) and superior performance on NuScenes-QA (59.5% accuracy).
5.The paper includes useful ablations on occupancy supervision and ego trajectory input.

**Weaknesses:**

1.The core architecture (cross-attention between occupancy queries and visual features) is relatively standard. The main contribution is using occupancy as supervision, which feels incremental.
2.The occupancy prediction results (~10% mIoU) are not comprehensively evaluated against specialized occupancy prediction methods
3.Several grammatical errors and awkward phrasings (e.g., "choozaizuose" in line 299)
4.The paper structure could be improved - the three-stage training is mentioned but not clearly motivated
5.All experiments are on nuScenes. How does the approach generalize to other datasets or driving scenarios?

**Questions:**

1.Can you provide actual inference time measurements comparing with/without occupancy prediction?
2.Can you ablate the value of λ_occ and provide more justification for your choice?

---

> ### Author Response · Authors · 2025-11-22
> **Strengthening OccVLA through Clarifications, Ablations, and Efficiency Analysis**
>
> # Response to Weakness
> We sincerely thank the reviewers for their insightful comments and constructive feedback. Below, we address each point in detail and provide additional experiments and clarifications.
> ## 1. Core architecture and contribution
> We acknowledge that our architecture is based on cross-attention between occupancy queries and visual features. Our main contribution lies in integrating the occupancy prediction process into the original transformer structure through trainable queries. This design ensures scalability and high compatibility with decoder-only language model frameworks.
> ## 2. Occupancy prediction performance and comparison
> Our occupancy mIoU (~10 %) is lower than specialized methods, but OccVLA is not specifically designed for occupancy. In typical occupancy prediction tasks, models usually take multi-frame image inputs, while OccVLA predicts occupancy from a single-frame input. This naturally introduces large invisible regions (e.g., voxels behind buildings or vehicles). Despite this, as shown in Figure 5 of our paper, OccVLA achieves accurate predictions for critical categories such as vehicles and drivable areas. This demonstrates that the model effectively captures essential 3D scene structures with the limited visual input.
> ## 3. Grammatical and typographical issues
> We sincerely apologize for the typographical error and grammatical issues. These will be corrected in the revised version of the paper.
> ## 4. Paper structure and motivation for the three-stage training
> The motivation for the three-stage training process is as follows:
> - Stage 1: Domain adaptation.
>   Since general-purpose VLMs are not not adequately trained on multi-view images, we first fine-tune the model on autonomous driving data to adapt it to multi-view (e.g., six-view) inputs and learn the domain-specific visual distribution of driving scenes.
> - Stage 2: Joint reasoning training.
>   After supervised fine-tuning (SFT), we simultaneously train meta-action reasoning and occupancy supervision to enhance planning and 3D understanding.
> - Stage 3: Planning head training.
>   The goal of this stage is to train a trajectory prediction head based on the meta-action outputs from the VLM, together with other features, to achieve accurate coordinate-level future trajectory planning.
> ## 5. Generalization beyond nuScenes
> All tests are currently conducted on nuScenes, but 3D occupancy is becoming a common voxel format with growing annotations and auto-labeling. Therefore, our proposed OccVLA framework can be easily adapted to other datasets or driving scenarios by incorporating occupancy supervision into existing VLM architectures, thereby enhancing general 3D understanding capabilities.
> # Response to Questions
> ## 1. Inference time comparison
> In response to the reviewers’ request, we evaluated the inference time with and without occupancy prediction. Specifically, we compared two configurations:
> (1) text generation based on both image tokens and occupancy tokens, and
> (2) text generation based only on image tokens, effectively skipping the occupancy prediction process.
> The results are as follows:
>
> | Text Access to Occupancy  | Speed (%) | Trajectory (%)   | Overall. (%) | Inference Time (s/it) |
> | ------------------------ | -------- | -------- | --------- | --------------------- |
> | ✓                          | 53.69  | 77.06 | 42.45    | 21.94                 |
> | ×                       | 54.83  | 77.24    | 43.08    | 17.95                 |
>
> Skipping occupancy prediction brings approximately a 25% speed improvement. Interestingly, bypassing occupancy prediction also yields a slight performance gain of about 0.5%. This suggests that an excessive number of visual tokens (occupancy tokens) may introduce redundant information that interferes with reasoning, leading to minor performance degradation. These results demonstrate that skipping the occupancy prediction process provides dual advantages in both inference efficiency and overall performance.
>
> ## 2. Ablation study on $λ_{occ}$
> We further conducted an ablation study to analyze the effect of the occupancy supervision weight $λ_{occ}$. The results are presented below:
>
> | $λ_{occ}$  |  Speed (%) | Trajectory (%)|  Overall.(%)   |
> | --------------- | -------- | --------- | ----- |
> | 0                 | 53.77  | 77.24     | 41.48 |
> | 0.1            | 53.85 | 75.77         | 41.45 |
> | 1               | 54.83  | 77.95       | 43.08 |
> | 2               | 54.01  | 74.07       | 40.64 |
>
> When $λ_{occ}$ = 0.1, occupancy supervision has little effect, similar to training without it. When $λ_{occ}$ = 2, the model overfits to occupancy and overall performance drops. Setting $λ_{occ}$ = 1 gives the best balance between using occupancy supervision and preventing overfitting.
>
> In summary, we appreciate the reviewers’ valuable feedback. We have added clarifications, ablations and timing data, and believe these additions strengthen the paper. Let us know if anything else is needed.

---

### Author Response · Authors · 2025-12-03
**Global Rebuttal Response**

Dear Reviewers and Area Chair,

We sincerely appreciate the time and effort you have dedicated to reviewing our manuscript.

We propose a new vision-language-action (VLA) framework for autonomous driving—**OccVLA**. OccVLA leverages **3D occupancy as an implicit supervision signal**, providing dense 3D guidance that effectively enhances the 3D scene understanding capability of VLMs/MLLMs.

As noted by the reviewers, this paper adopts occupancy as implicit supervision, which has **a creative and strongly motivated design** (5Rfz, QBwy, E7ic). (5Rfz, QBwy, E7ic), proposes a **computationally efficient approach** (5Rfz, ho2c), and is supported by **comprehensive empirical results** on **trajectory planning and VQA tasks** in **nuScenes** (5Rfz, QBwy, ho2c).

We are very grateful for the reviewers’ constructive feedback. In response, we have carefully revised and improved the manuscript as follows:

- **Ablation studies** on `λ_occ` and **COT supervision** (Appendix A)
- **Detailed explanation** of the conditional dependency in occupancy and **efficiency comparison** of the skip mechanism (Appendix B)
- **Extended experiments** on the **Waymo Open Dataset End-to-End Driving (WOD-E2E)** to evaluate the **generalization of OccVLA in long-tail scenarios** (Appendix C)
- **More rigorous discussion** on the comparison between **occupancy and 3D bounding boxes**: while 3D bounding boxes provide only limited and category-dependent spatial constraints, serving as sparse supervision for VLMs, occupancy enables **fine-grained, open-world 3D scene modeling** and offers **dense supervision** based on the method proposed in OccVLA (with detailed explanations included in the revision)
- **Fixed typographical errors**

In the revised manuscript, these updates are temporarily **highlighted in blue** for your convenience.

We sincerely hope that these updates will help more clearly convey the advantages and contributions of **OccVLA** to the ICLR community.

Thank you very much,
**The Authors**

---

### Meta-Review · Area_Chair_CQ7W · 2026-01-09

**Summary:**

Authors propose a new vision-language-action (VLA) framework for autonomous driving—OccVLA. OccVLA leverages 3D occupancy as an implicit supervision signal, providing dense 3D guidance that effectively enhances the 3D scene understanding capability of VLMs/MLLMs. OccVLA achieves state-of-the-art results on the nuScenes benchmark for trajectory planning and demonstrates superior performance on 3D visual question-answering tasks.

**Reviewer Concerns:**

Concerns from four reviewers mainly centered around:

- Network architecture is trivial.
- Prediction results are not evaluated in great details. / Single dataset evaluation (nuScenes)
- Invalid claim regarding existing VLM supervision
- The argument against the scalability of 3D bounding box supervision is weakened by recent advances in auto-labeling.
- The characterization of the EMMA baseline is inaccurate and potentially misleading.
- Lack of related work.
- Significant resemblance to previous work
- Technical details / lack of experiments

**Reviewer Scores:**

Authors did a comprehensive job addressing most concerns by reviewers, including:

+ explaining the novelty / distinction from previous work
+ adding more experiments on the new benchmark, WOD-E2E
+ adding more ablations on the analysis of different components.

AC read the paper, rebuttal, review comment in great details. Still the paper bears limitted technical contribution compared to previous works (e.g. Ross) and marginal experimental improvement over previous methods (as shown in Table 1).

The key distinction is the objective "3D spatial perception rather than reconstructing visual content", as authors wrote, is not distinctive as it is in AC's perspective.

---

### Decision · Program_Chairs · 2026-01-26

Reject